# Learning Compositional Behaviors from Demonstration and Language

**Weiyu Liu**[1*], **Neil Nie**[1*], **Ruohan Zhang**[1], **Jiayuan Mao**[2†], **Jiajun Wu**[1†]

[1]Stanford University    [2]MIT

**Abstract:** We introduce Behavior from Language and Demonstration (BLADE), a framework for long-horizon robotic manipulation by integrating imitation learning and model-based planning. BLADE leverages language-annotated demonstrations, extracts abstract action knowledge from large language models (LLMs), and constructs a library of structured, high-level action representations. These representations include preconditions and effects grounded in visual perception for each high-level action, along with corresponding controllers implemented as neural network-based policies. BLADE can recover such structured representations automatically, without manually labeled states or symbolic definitions. BLADE shows significant capabilities in generalizing to novel situations, including novel initial states, external state perturbations, and novel goals. We validate the effectiveness of our approach both in simulation and on a real robot with a diverse set of objects with articulated parts, partial observability, and geometric constraints.

**Keywords:** Manipulation, Planning Abstractions, Learning from Language

## 1 Introduction

Developing autonomous robots capable of completing long-horizon manipulation tasks is a significant milestone. We want to build robots that can directly perceive the world, operate over extended periods, generalize to various states and goals, and are robust to perturbations. A promising direction is to combine learned policies with model-based planners, allowing them to operate on different time scales. In particular, imitation learning-based methods have proven highly successful in learning policies for various "behaviors," which usually operate over a short time span [e.g., 1]. To solve more complex and longer-horizon tasks, we can compose these behaviors by planning in abstract action spaces [2–4], in latent spaces [5], or via large pre-trained models such as large language models [6].

However, one of the key challenges of all high-level planning approaches is the automatic acquisition of an abstraction for the learned "behaviors" to support long-horizon planning. The goal of this behavior abstraction learning is to build representations that describe the preconditions and effects of behaviors, to enable chaining and search. These representations should depend on the environment, the set of possible goals, and the specifications of individual behaviors. Furthermore, these representations should be grounded on high-dimensional perception inputs and low-level robot control commands.

Our insight into tackling this challenge is to leverage knowledge from two sources: the low-level, mechanical understanding of robot-object contact, and the high-level, abstract understanding of object-object interactions described in language that can be extracted from language models as the knowledge source. Our framework, behavior from language and demonstration (BLADE), takes as input a small number of language-annotated demonstrations (Fig. 1a). It segments each trajectory based on which object is in contact with the robot. Then, it uses a large language model (LLM), conditioned on the contact sequences and the language annotations, to propose abstract behavior descriptions with preconditions and effects that best explain the demonstration trajectories. During training, we extract the state abstraction terms from the preconditions and effects (e.g., *turned-on*,

---

[*] denotes equal contribution. [†] denotes equal advising. Project page and videos: https://blade-bot.github.io/.

8th Conference on Robot Learning (CoRL 2024), Munich, Germany.

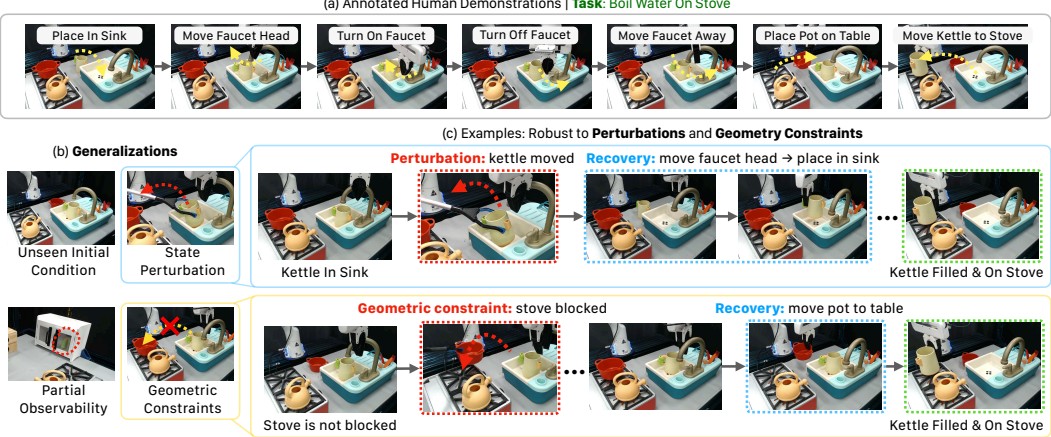

**Figure 1:** BLADE, a robot manipulation framework combining imitation learning and model-based planning. (a) BLADE takes language-annotated demonstrations as training data. (b) It generalizes to unseen initial conditions, state perturbations, and geometric constraints. (c) In the depicted scenarios, BLADE recovers from perturbations such as moving the kettle out of the sink, and resolves geometric constraints including a blocked stove.

*aligned-with*), and learn their groundings on perception inputs. We also learn the control policies associated with each behavior (e.g., *turn on the faucet*).

Our model offers several advantages. First, unlike prior work that relies on manually defined state abstractions or additional state labels, our method automatically generates state abstraction labels based on the language-annotated demonstrations and LLM-proposed behavior descriptions. BLADE recovers the visual grounding of these abstractions without any additional label. Second, BLADE generalizes to novel states and goals by composing learned behaviors using a planner. Shown in Fig. 1b, it can handle various novel initial conditions and external perturbations that lead to unseen states. Third, our method can handle novel geometric constraints (Fig. 1c) and partial observability from articulated bodies like drawers.

## 2 Related Work

**Composing skills for long-horizon manipulation.** A large body of model-based planning methods use manually-defined transition models [2, 7–12] or models learned from data [13–18] to generate long-horizon plans. However, learning dynamics models with accurate long-term predictions and strong generalization remains challenging. A related direction is to introduce hierarchical structures into the policy models [19–25], where different methods can segment continuous demonstrations into short-horizon skills [23, 26, 27]. Facing the challenges in modeling action dependencies, these methods are limited to following sequentially specified subgoals. Some work addresses this issue by learning the dependencies between actions from data, but they require large-scale supervised datasets [28–31]. Our approach is related to methods that learn symbolic action representations [32–36]; the difference is that BLADE uses a LLM to generates causal models of the environment and learns their groundings on sensory inputs.

**Using LLMs for planning.** Many researchers have explored using LLMs for planning. Methods for direct generation of action sequences [37, 38] can struggle to produce accurate plans [39, 40]. Researchers have also leveraged LLMs as translators from natural language instructions to symbolic goals [41–44], as generalized solvers [45], as memory modules [46], and as world models [47, 48]. To improve the planning accuracy, prior work has explored techniques including using programs [49, 50], learning affordance functions [6, 51], replanning [52], finetuning [53–55], embedding reasoning in a behavior tree [56], and VLM-based decision-making [57, 58]. BLADE shares a similar spirit as methods using LLMs to generate planning-compatible action representations [59–61]. However, they make assumptions on the availability of state abstractions, while BLADE grounds LLM-generated action definitions without additional labels. Also complementary to methods that leverage these representations for skill learning [62, 63], our approach uses them for composing skills in novel ways.

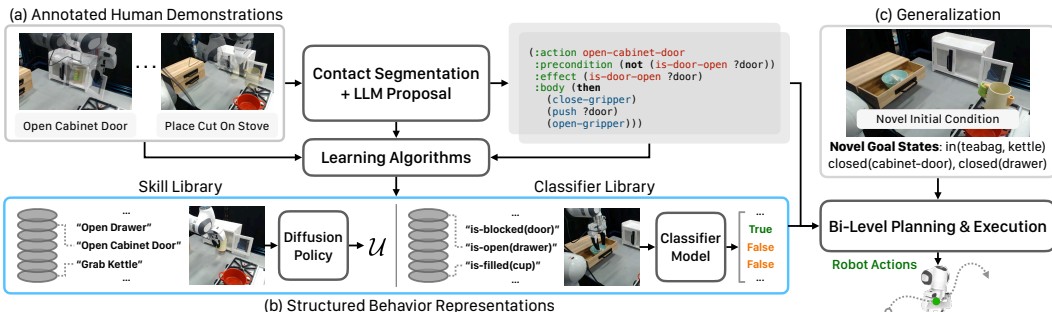

**Figure 2: Overview of BLADE.** (a) BLADE receives language-annotated human demonstrations, (b) segments demonstrations into contact primitives, and learns a structured behavior representation. (c) It generalizes to novel conditions by leveraging bi-level planning and execution to achieve goal states.

# 3 Problem Formulation

We consider the problem of learning a language-conditioned goal-reaching manipulation policy. Formally, the environment is modeled as a tuple $\langle \mathcal{X}, \mathcal{U}, \mathcal{T} \rangle$ where $\mathcal{X}$ is the raw state space, $\mathcal{U}$ is the low-level action space, and $\mathcal{T} : \mathcal{X} \times \mathcal{U} \to \mathcal{X}$ is the transition function (which may be stochastic and unknown). Furthermore, the robot will receive observations $o \in \mathcal{O}$ that may be partially observable views of the states. At test time, the robot also receives a natural language instruction $\ell_t$, which corresponds to a set of goal states. An oracle goal satisfaction function defines whether the language goal is reached, i.e., $g_{\ell_t} : \mathcal{X} \to \{T, F\}$. Given an initial state $x_0 \in \mathcal{X}$ and the instruction $\ell_t$, the robot should generate a sequence of low-level actions $\{u_1, u_2, ..., u_H\} \in \mathcal{U}^H$.

In the language-annotated learning setting, the robot has a dataset of language-annotated demonstrations $\mathcal{D}$. Each demonstration is a sequence of robot actions $\{u_1, ..., u_H\}$ paired with observations $\{o_0, ..., o_H\}$. Each trajectory is segmented into $M$ sub-trajectories, and natural language descriptions $\{\ell_1, ..., \ell_M\}$ are associated with the segments (e.g., "*place the kettle on the stove*"). In this paper, we assume that there is a finite number of possible $\ell$'s—each corresponding to a skill to learn.

Directly learning a single goal-conditioned policy that can generalize to novel states and goals is challenging. Therefore, we recover an *abstract* state and action representation of the environment and combine online planning in abstract states and offline policy learning for low-level control to solve the task. In BLADE, behaviors are represented as temporally extended actions with preconditions and effects characterized by state predicates. Formally, we want to recover a set of predicates $\mathcal{P}$ that define an abstract state space $\mathcal{S}$. We focus on a scenario where all predicates are binary. However, they are grounded on high-dimensional sensory inputs. Using $\mathcal{P}$, a state can be described as a set of grounded atoms such as $\{kettle(\mathrm{A}), stove(\mathrm{B}), filled(\mathrm{A}), on(\mathrm{A}, \mathrm{B})\}$ for a two-object scene. BLADE will learn a function $\Phi : \mathcal{O} \to \mathcal{S}$ that maps observations to abstract states. In its current implementation, BLADE requires humans to additionally provide a list of predicate names in natural language, which we have found to be helpful for LLMs to generate action definitions. We provide additional ablations in the Appendix A.2. Based on $\mathcal{S}$, we learn a library of *behaviors* (a.k.a., *abstract actions*). Each behavior $a \in \mathcal{A}$ is a tuple of $\langle name, args, pre, eff, \pi \rangle$. *name* is the name of the action. *args* is a list of variables related to the action, often denoted by $?x, ?y$. *pre* and *eff* are the precondition and effect formula defined in terms of the variables *args* and the predicates $\mathcal{P}$. A low-level policy $\pi : \mathcal{O} \to \mathcal{U}$ is also associated with $a$. The semantics of the preconditions and effects is: for any state $x$ such that $pre(\Phi(x))$ is satisfied, executing $\pi$ at $x$ will lead to a state $x'$ such that $eff(\Phi(x'))$ [64].

# 4 Behavior from Language and Demonstration

BLADE is a method for learning abstract state and action representations from language-annotated demonstrations. It works in three steps, as illustrated in Fig. 2. First, we generate a symbolic behavior definition conditioned on the language annotations and contact sequences in the demonstration using a large language model (LLM). Next, we learn the classifiers associated with all state predicates and the control policies, all from the demonstration without additional annotations. At test time, we use a bi-level planning and execution strategy to generate robot actions.

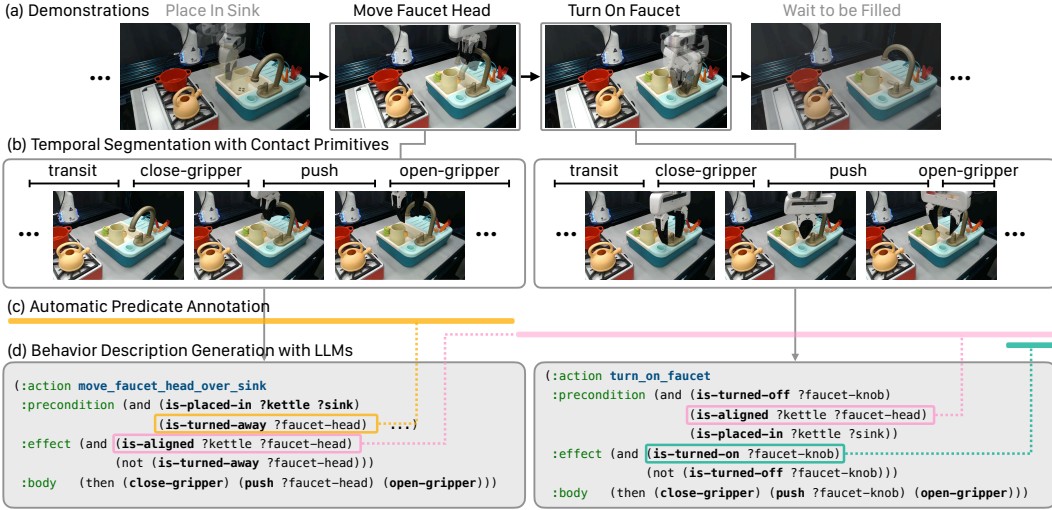

**Figure 3: Behavior Descriptions Learning.** (a) A demonstration is provided along with corresponding language annotations. (b) The demonstration is segmented into a sequence of contact primitives. (c) A large language model interprets the annotation and contact sequence, generating a symbolic behavior definition. (d) The system automatically generates data to learn classifiers for state predicates.

## 4.1 Behavior Description Learning

Given a finite set of behaviors with language descriptions $\{\ell\}$ and corresponding demonstration segments, we generate an abstract description for each $\ell$ by querying large language models. To facilitate LLM generation, we provide additional information on the list of objects with which the robot has contact. The generated operators are further refined with abstract verification.

**Temporal segmentation.** We first segment each demonstration (Fig. 3a) into a sequence of *contact-based primitives* (Fig. 3b). In this paper we consider seven primitives describing the interactions between the robot and other objects: *open/close* grippers without holding objects, *move-to*$(x)$ which moves the gripper to an object, *grasp*$(x, y)$ and *place*$(x, y)$ which grasp and place object $x$ from/onto another object $y$, *move*$(x)$ which moves the currently holding object $x$ and *push*$(x)$. We leverage proprioception, i.e., gripper open state, and object segmentation to automatically segment the continuous trajectories into these basis segments. For example, pushing the faucet head away involves the sequence of $\{close\text{-}gripper, push, open\text{-}gripper\}$. This segmentation will be used for LLMs to generate operator definitions and for constructing training data for control policies.

**Behavior description generation with LLMs.** Our behavior description language is based on PDDL [65]. We extend the PDDL definition to include a *body* section which is a sequence of contact primitives. It will be generated by the LLM based on the demonstration data.

Our input to the LLM mainly contains: 1) a general description of the environment, 2) the natural language descriptions $\ell$ associated with the behavior itself and other behaviors that have appeared preceding or following $\ell$ in the dataset, 3) all possible sequence of contact primitive sequences associated with $\ell$ across the dataset, and 4) additional instructions on the PDDL syntax, including a single PDDL definition example. We find the additional context useful. As shown in Fig. 3d, in addition to preconditions and effects of the operators, we also ask LLMs to predict a *body* of contact primitive sequence associated with the behavior, which we call *body*. We assume that each behavior has a single corresponding contact primitive sequence, and use this step to account for noises in the segmentation annotations. After LLM predicts the definition for all behavior, we will re-segment the demonstrations associated with each behavior based on the LLM-predicted body section.

**Behavior description refinement with abstract verification.** In addition to checking for syntax errors, we also verify the generated behavior descriptions with *abstract verification* on the demonstration trajectories. Given a segmented sequence of the trajectory where each segment is associated with a behavior, we verify whether the preconditions of each behavior can be satisfied by the accumulated effects of the previous segments. This verification does not require learning the grounding of

predicates and can be done at the behavior level for incorrect preconditions and effects, and at the contact primitive level for missing or incorrect contact primitives (e.g., *grasp* cannot be immediately followed by other *grasp*). We resample behavior definitions that do not pass the verification.

## 4.2 Classifier and Policy Learning

Given the dataset of state-action segments associated with each behavior, we train the classifiers for different state predicates and the low-level controller for each behavior.

**Automatic predicate annotation.** We leverage *all* behavior descriptions to automatically label an observation sequence $\{o_1, ..., o_H\}$ based on its associated segmentation. In particular, at $o_0$, we label all state predicates as "unknown." Next, we unroll the sequence of behavior executed in $\bar{o}$. As illustrated in Fig. 3c, before applying a behavior $a$ at step $o_t$, we label all predicates in $pre_a$ true and predicates in $eff_a$ false. When $a$ finishes at step $o_{t'}$, we label all predicates in $eff_a$. In addition, we will propagate the labels for state predicates to later time steps until they are explicitly altered by another behavior $a$. In contrast to earlier methods, such as Migimatsu and Bohg [66] and Mao et al. [67], which directly use the first and last state of state-action segments to train predicate classifiers, our method greatly increases the diversity of training data. After this step, for each predicate $p \in \mathcal{P}$, we obtain a dataset of paired observations $o$ and the predicate value of $p$ at the corresponding time step.

**Classifier learning.** Based on the state predicate dataset generated from behavior definitions, we train a set of state classifiers $f_\theta(p) : \mathcal{O} \to \{T, F\}$, which are implemented as standard neural networks for classification. We include implementation details in Appendix A.6. In real-world environments with strong data-efficiency requirements, we additionally use an open vocabulary object detector [68] to detect relevant objects for the state predicate and crop the observation images. For example, only pixels associated with the object faucet will be the input to the *turned-on*(faucet) classifier.

**Policy learning.** For each behavior, we also train control policies $\pi_\theta(a) : \mathcal{O} \to \mathcal{U}$, implemented as a diffusion policy [1]. In simulation, we use a combination of frame-mounted and wrist-mounted RGB-D cameras as the inputs to the diffusion policy, while in the real world, the policy takes raw camera images as input. The high-level planner orchestrates which of these low-level policies to deploy based on the scene and states. Once trained on these diverse demonstrations of different skills, the resulting low-level policies can adapt to local changes, such as variations in object poses.

## 4.3 Bi-Level Planning and Execution

At test time, given a novel state and a novel goal, BLADE first uses LLMs to translate the goal into a first-order logic formula based on the state predicates. Next, it leverages the learned state abstractions to perform planning in a symbolic space to produce a sequence of behaviors. Then, we execute the low-level policy associated with the first behavior, and we re-run the planner after the low-level policy finishes—this enables us to handle various types of uncertainties and perturbations, including execution failure, partial observability, and human perturbation. In implementation, we use the fast-forward heuristic to generate plans [69]; however, our method is planner-agnostic, and other symbolic planners (e.g., Fast-Downward [70]) are compatible.

Visibility and geometric constraints are also modeled as preconditions, in addition to other object-state and relational conditions. For example, the behavior "opening the cabinet door" will have preconditions on the initial state of the door, a visibility constraint that the door is visible, and a geometric constraint that nothing is blocking the door. When those preconditions are not satisfied, the planner will automatically generate plans, such as actions that move obstacles away, to achieve them. Partial observability was handled by using the most-likely state assumption during planning and performing replanning. We include details in Appendix A.8.

## 5 Experiments

### 5.1 Simulation Experimental Setup

We use the CALVIN benchmark [71] for simulation-based evaluations, which include teleoperated human-play data. We use the split $D$ of the dataset, which consists of approximately 6 hours of interactions. Annotations of the play data are generated by a script that detects goal conditions

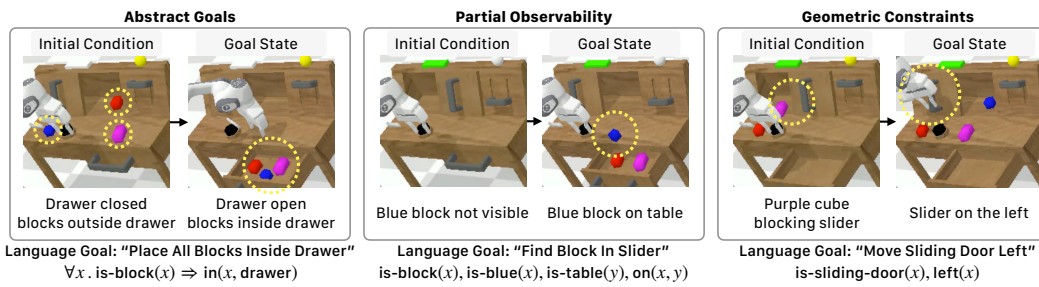

**Figure 4: Generalization Tasks in CALVIN.** Examples from the three generalization tasks in the CALVIN simulation environment. Successfully completing these tasks require planning for and executing 3-7 actions.

**Table 1: Generalization results in CALVIN.** Mean success rates with STD from three seeds are reported. BLADE outperforms latent planning, LLM, and VLM baselines in completing novel long-horizon tasks.

| Method | State Classifier | Latent Feasibility | Generalization Task | | |
|---|---|---|---|---|---|
| | | | Abstract Goal | Geometric Constraint | Partial Observability |
| HULC [72] | N/A | N/A | $2.78 \pm 3.47$ | $11.67 \pm 11.55$ | $0.00 \pm 0.00$ |
| SayCan [6] | N/A | Short | $23.89 \pm 1.92$ | $1.67 \pm 2.89$ | $1.67 \pm 2.89$ |
| VILA [57] | N/A | N/A | $18.38 \pm 2.48$ | $0.00 \pm 0.00$ | $4.17 \pm 5.20$ |
| T2M-Shooting [51] | Learned | Long | $57.78 \pm 12.29$ | $0.00 \pm 0.00$ | $13.33 \pm 1.44$ |
| Ours | Learned | N/A | $\mathbf{68.33 \pm 10.14}$ | $\mathbf{26.67 \pm 7.64}$ | $\mathbf{75.83 \pm 3.82}$ |
| T2M-Shooting [51] | GT | Long | $61.67 \pm 5.00$ | $0.00 \pm 0.00$ | $0.83 \pm 1.44$ |
| Ours | GT | N/A | $\mathbf{76.11 \pm 6.74}$ | $\mathbf{56.67 \pm 16.07}$ | $\mathbf{70.00 \pm 5.00}$ |

on simulator states, and there are in total 34 types of behaviors. We use RGB-D images from the mounted camera for classifier learning and partial 3D point clouds recovered from the images for policy learning. The original benchmark focuses only on evaluating individual skills and instruction following. To evaluate the ability to compositionally combine previously learned policies to solve novel tasks, we design six new generalization tasks, with examples shown in Fig. 4. Each task has a language instruction, a sampler that generates random initial states, and a goal satisfaction function for evaluation. For each task, we sample 20 initial states and evaluate all methods with three different random seeds. See Appendix B.1 for more details on the benchmark setup.

**Baselines.** We compare BLADE with two groups of baselines: hierarchical policies with planning in latent spaces and LLM/VLM-based methods for robotic planning. For the former, we use HULC [72], a representative method in CALVIN, which learns a hierarchical policy from language-annotated play data using hindsight labeling. For the latter, we use SayCan [6], Robot-VILA [57], and Text2Motion [51]. Note that Text2Motion assumes access to ground-truth symbolic states. Hence we compare Text2Motion with BLADE in two settings: one with the ground-truth states and the other with the state classifiers learned by BLADE. See Appendix B.2 for more details on these methods.

### 5.2 Results in Simulation

Table 1 presents the performance of different models in all three types of generalization tasks.

**Structured behavior representations improve long-horizon planning.** We first compare to the hierarchical policy HULC in Table. 1. BLADE with learned classifiers achieves a more than 65% improvement in the success rate for reaching abstract goals while using the same language-annotated play data. We attribute this to the particular implementation of hindsight labeling in HULC being not sufficient to generate plans that chain multiple high-level actions: for example, the task of placing all blocks in the closed drawer requires chaining together a minimum of 7 behaviors.

**Structured transition models learned by BLADE facilitate long-horizon planning.** Both SayCan and T2M-Shooting uses learned action feasibility models for planning. Shown in Table. 1, learning accurate feasibility models directly from raw demonstration data remains a significant challenge. In our experiment, we find that first, when the LLM does not take into account state information (SayCan), using the short-horizon feasibility model is not sufficient to produce sound plans. Second, since our model learns a structured transition model, factorized into different state predicates, BLADE

is capable of producing more accurate longer-horizon plans than T2M-Shooting which learns long-horizon feasibility from data.

**Structured scene representations facilitate making feasible plans.** Compared to the Robot-VILA method, which directly predicts action sequences based on the image state, BLADE first uses learned state classifiers to construct an abstract state representation. This contributes to a 49% improvement on the Abstract Goal tasks in Table 1. We observe that the pre-trained VLM used in Robot-VILA often predicts actions that are not feasible in the current state. For example, Robot-VILA consistently performs better in completing "placing all blocks in a closed drawer" than "placing all blocks in an open drawer" since it always predicts opening the drawer as the first step.

**Explicit modeling of geometric constraints and object visibility improves performance in these scenarios.** BLADE can reason about these challenging situations without explicitly being trained in those settings. Table. 1 shows that our approach consistently outperforms baselines in these two settings. These generalization capabilities are built on the explicit modeling of geometric constraints and object visibility in behavior preconditions.

**BLADE can automatically propose operators for the specific environment given demonstrations.** Our experiment shows that the LLM can automatically propose high-quality behavior descriptions that resemble the dependency structures among operators. For example, the LLM discovers from the given contact primitive sequences and language-paired demonstration that blocks can only be placed after the block is lifted and that a drawer needs to be opened before placing objects inside, etc. Some of these dependencies are unique to the CALVIN environment, therefore requiring the LLM to generate specifically for this domain. We provide more visualizations in the Appendix A.1.

**BLADE's automatic predicate annotation enables better classifier learning.** From Table 1, we observe that having accurate state classifier models is critical for algorithms' performance (GT vs. Learned). Hence, we perform additional ablation studies on classifier learning. Prior work such

**Table 2:** Ablation on state classifier learning in CALVIN.

| Method | Abstract | Geometric | Partial Obs. |
|--------|----------|-----------|--------------|
| [66]   | $33.89 \pm 5.85$ | $9.17 \pm 5.20$ | $3.33 \pm 2.89$ |
| BLADE  | $\mathbf{68.33 \pm 10.14}$ | $\mathbf{26.67 \pm 7.64}$ | $\mathbf{75.83 \pm 3.82}$ |

as Migimatsu and Bohg [66] also presented a method for learning the preconditions and effects of actions from segmented trajectories and symbolic action descriptions. The key difference between BLADE and theirs is that they only use the first and last frame of each segment to supervise the learning of state classifiers. We compare the two classifier learning algorithms, given the same LLM-generated behavior definitions, by evaluating the classifier accuracy on held-out states. BLADE shows a 20.7% improvement in F1 (16.3% improvement for classifying object states and 38.6% improvement for classifying spatial relations) compared to the baseline model. This also translates into significant improvements in the planning success rate, as shown in Table 2.

### 5.3 Real World Experiments

**Environments.** We use a Franka Emika robot arm with a parallel jaw gripper. The setup includes five RealSense RGB-D cameras, with one being wrist-mounted on the robot and the remaining positioned around the workspace. Fig. 5 shows the two domains: Make Tea and Boil Water. For each domain, we collect 85 language-annotated demonstrations using teleoperation with a 3D mouse. After segmenting the demonstrations using proprioception sensor data, an LLM is used to generate behavior descriptions. These descriptions are subsequently used for policy and classifier learning.

**Setup.** We compare BLADE against the VLM-based baseline Robot-VILA. We omit SayCan and T2M-Shooting since they require additional training data. We first test the original action sequences seen in the demonstrations for each domain. We then test on tasks that require novel compositions of behaviors for four types of generalizations, i.e., unseen initial condition, state perturbation, geometric constraints, and partial observability. For each generalization type, we run six experiments and report the number of experiments that have been successfully completed. See Appendix D for details.

**Results.** In Fig. 5, we show that our model is able to successfully complete at least 4/6 tasks for all generalization types in the two different domains. In comparison, Robot-VILA struggles to generate

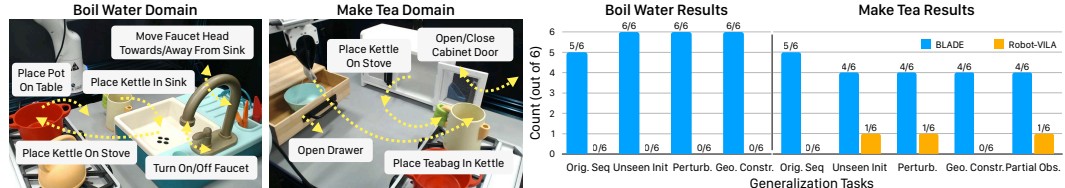

**Figure 5: Domains and Results in Real World. Make Tea** features a toy kitchen designed to simulate boiling water on a stove. The robot must assess the available space on the stove for the kettle. It also needs to manage the dependencies between actions, such as the faucet must be turned away before the kettle can be placed into the sink to avoid collisions. **Boil Water** involves a tabletop task aimed at preparing tea, incorporating a cabinet, a drawer, and a stove. The robot must locate the kettle, potentially hidden within the cabinet, and a teabag in the drawer. Additionally, it must consider geometric constraints by removing obstacles that block the cabinet doors. In both environments, our model significantly outperforms the VLM-based planner Robot-VILA.

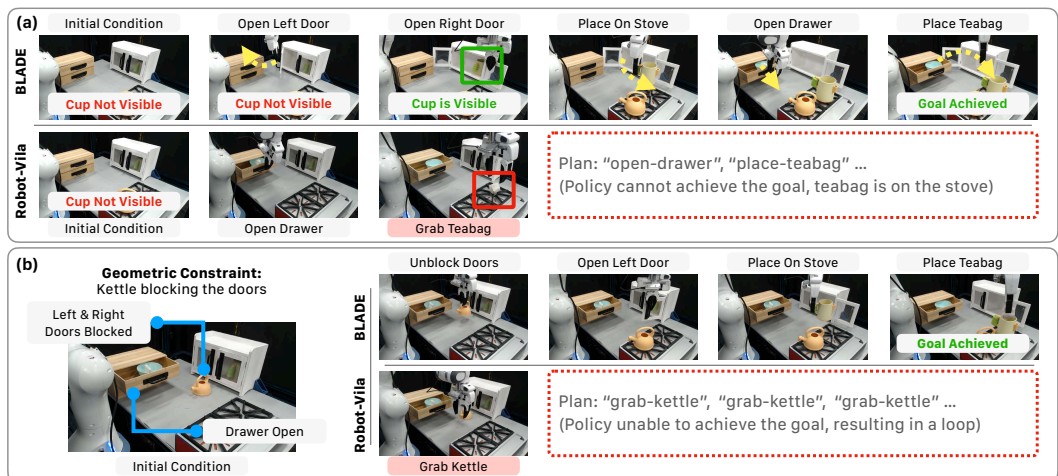

**Figure 6: Real World Planning and Execution.** We show the execution traces from BLADE and Robot-VILA for two generalization tasks: (a) partial observability and (b) geometric constraints.

correct plans to complete the tasks. In Fig. 6, we visualize the generated plans and execution traces of both models. In example (a), we show that BLADE can find the kettle initially hidden in the cabinet and then complete the rest of the task. In comparison, Robot-VILA directly predicts placing the teabag in the kettle when the kettle is not visible, resulting in a failure.

## 6 Conclusion and Discussion

BLADE is a novel framework for long-horizon manipulation by integrating model-based planning and imitation learning. BLADE uses an LLM to generate behavior descriptions with preconditions and effects from language-annotated demonstrations and automatically generates state abstraction labels based on behavior descriptions for learning state classifiers. At performance time, BLADE generalizes to novel states and goals by composing learned behaviors with a planner. Compared to latent-space and LLM/VLM-based planners, BLADE successfully completes significantly more long-horizon tasks with various types of generalizations.

**Limitations.** One limitation of BLADE is that the automatic segmentation of demonstrations is based on gripper states; more advanced contact detection techniques might be required for certain tasks such as caging grasps. We also assume the knowledge of a given set of predicate names in natural language and focus on learning dependencies between actions using the given predicates. Automatically inventing task-specific predicates from demonstrations and language annotations, possibly with the integration of vision-language models (VLMs) is an important future direction. In our experiments, we also found that noisy state classification led to some planning failures. Therefore, developing planners that are more robust to noises in state estimation is necessary. Finally, achieving novel compositions of behaviors also requires policies with strong generalization to novel environmental states, which remain a challenge for skills learned from a limited amount of demonstration data.

**Acknowledgments**

This work is in part supported by Analog Devices, MIT Quest for Intelligence, MIT-IBM Watson AI Lab, ONR Science of AI, NSF grant 2214177, ONR N00014-23-1-2355, AFOSR YIP FA9550-23-1-0127, AFOSR grant FA9550-22-1-0249, ONR MURI N00014-22-1-2740, ARO grant W911NF-23-1-0034. We extend our gratitude to Jonathan Yedidia, Nicholas Moran, Zhutian Yang, Manling Li, Joy Hsu, Stephen Tian, Chen Wang, Wenlong Wang, Yunfan Jiang, Chengshu Li, Josiah Wong, Mengdi Xu, Sanjana Srivastava, Yunong Liu, Tianyuan Dai, Wensi Ai, Yihe Tang, the members of the Stanford Vision and Learning Lab, and the anonymous reviewers for insightful discussions.

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

# Supplementary Material for Learning Compositional Behaviors from Demonstration and Language

This supplementary material provides additional details on the BLADE framework, the experiments, and qualitative examples. Section A provides a detailed description of the method, including the behavior description generation, predicate generation, abstract verification, automatic predicate annotation, classifier implementation, and policy implementation. Section B provides details on the simulation experiments, including the task design and baseline implementations. Section C provides qualitative examples of our method and baselines. Section D provides details of our setup of the real-robot experiment. Finally, Section E includes a full list of the prompts for the baselines used in the simulation experiments.

## A   BLADE Details

### A.1   Behavior Description Generation with LLMs

In Listing 1, we show the behavior descriptions automatically generated by the LLM for the CALVIN domain. We also show the detailed prompt to the LLM for generating the behavior description. We break down the system prompt into four parts: definitions of primitive actions (Listings 2), definitions of predicates and environment context (Listings 3), an in-context example (Listings 4), and additional instructions (Listings 5). In Listings 6, we show one example of the specific user prompt that is used to generate the behavior description for *place-in-drawer*.

In our experiments, we find that the environment description is necessary for the LLMs to understand the context of the task. For the simulation experiment, we provide the environment description as a list of objects and brief explanations, as shown in Listings 3. However, this description can be automatically generated using image tagging models such as Recognize Anything [73] or general-purpose VLMs such as GPT-4V.

### A.2   Predicate Generation with LLMs

Our algorithms are agnostic to the source of predicates and can flexibly generate action descriptions based on the given predicates. In our main experiment, we assume that the predicates for each task domain are provided in natural language. Here, we show that given the task definition and the environment context, a LLM can automatically generate the relevant predicates for the domain. Then, we generate behavior descriptions based on the automatically generated predicates.

To generate high-quality predicates and behavior descriptions, we take the following steps. First, the LLM is provided with the list of objects in the scene and the language-paired demonstration sequence and is required to generate relevant predicates for the domain. In Listing 7, we show an example of the input prompt to the LLM for the CALVIN domain. Second, the LLM is prompted with the generated predicates and is required to ground the predicates using the objects in the scene. This step helps the LLM eliminate errors in the first step, such as missing arguments. Finally, we instruct the LLM to remove semantically equivalent predicates and keep the most general ones. For example, *is-on-stove*(kettle) is removed by the LLM in our experiment because *is-on*(kettle, stove) is semantically equivalent. After the predicates are automatically generated and filtered, we proceed with the behavior description generation.

In Table A1, Table A2, and Table A3, we compare the generated predicates with the predicates defined by the domain expert for the CALVIN, Boil Water, and Make Tea domains. We observe that the LLM is able to generate 28 out of 30 predicates that match closely with the expert-designed predicates. These predicates provide abstract representations for object states (e.g., *is-open*, *light-on*), relations between objects (e.g., *in-slider*, *in*), and robot-centric states (e.g., *holding*). The LLM incorrectly generates the predicate *next-to*($?x, ?y$) to characterize the effects of the *push-left* and *push-right* actions, possibly due to ambiguities in the definition of the actions. For the two real-world domains,

we perform additional experiments to confirm that the generated predicates and behavior descriptions can be used to generate correct task plans on all nine real-world tasks presented in Section 5.3.

**Table A1: Comparison of Manually Defined and Automatically Generated Predicates for CALVIN.**

| Manually Defined | Automatically Generated |
| --- | --- |
| *rotated-left*(?x) | *rotated-left*(?x) |
| *rotated-right*(?x) | *rotated-right*(?x) |
| *lifted*(?x) | *holding*(?x) |
| *is-open*(?x) | *is-open*(?x) |
| *is-close*(?x) | *is-closed*(?x) |
| *is-turned-on*(?x) | *light-on*(?x) |
| *is-turned-off*(?x) | *light-off*(?x) |
| *is-slider-left*(?x) | *slider-left*(?x) |
| *is-slider-right*(?x) | *slider-right*(?x) |
| *is-on*(?x, ?y) | *on-table*(?x) |
| *is-in*(?x, ?y) | *in-slider*(?x), *in-drawer*(?x) |
| *stacked*(?x, ?y) | *on*(?x, ?y) |
| *unstacked*(?x, ?y) | *clear*(?x) |
| *pushed-left*(?x) | - |
| *pushed-right*(?x) | - |
| - | *next-to*(?x, ?y) |

**Table A2: Comparison of Manually Defined and Automatically Generated Predicates for Boil Water.**

| Manually Defined | Automatically Generated | Automatically Generated (Grounded) |
| --- | --- | --- |
| *is-placed-on*(?x, ?y) | *on-table*(?x) | *on-table*(kettle), *on-table*(pot) |
| *is-placed-in*(?x, ?y) | *in-sink*(?x) | *in-sink*(kettle), *in-sink*(pot) |
| *is-blocked*(?x) | *on-stove*(?x) | *on-stove*(pot) |
| *is-turned-away*(?x) | *faucet-over-sink*(?x, ?y) | *faucet-over-sink*(faucet, sink) |
| *is-aligned*(?x, ?y) | *faucet-over-sink*(?x, ?y) | *faucet-over-sink*(faucet, sink) |
| *is-turned-on*(?x) | *faucet-on*(?x) | *faucet-on*(faucet) |
| *is-turned-off*(?x) | *faucet-off*(?x) | *faucet-off*(faucet) |
| *is-filled*(?x) | *filled*(?x) | *filled*(kettle), *filled*(pot) |
| *holding*(?x) | *holding*(?x) | *holding*(pot) |

**Table A3: Comparison of Manually Defined and Automatically Generated Predicates for Make Tea.**

| Manually Defined | Automatically Generated | Automatically Generated (Grounded) |
| --- | --- | --- |
| *is-placed-on*(?x, ?y) | *is-on*(?x, ?y) | *is-on*(kettle, stove) |
| *is-cabinet-door-open*(?x) | *is-open*(?x) | *is-open*(left-door), *is-open*(right-door) |
| *is-placed-inside*(?x, ?y) | *is-in-drawer*(?x) | *is-in-drawer*(teabag) |
| | *is-in-kettle*(?x) | *is-in-kettle*(teabag) |
| | *is-in-cabinet-left*(?x) | *is-in-cabinet-left*(kettle) |
| | *is-in-cabinet-right*(?x) | *is-in-cabinet-right*(kettle) |
| *is-drawer-open*(?x) | *is-open*(?x) | *is-open*(drawer) |
| *is-left-cabinet-door-blocked*(?x) | *is-blocking*(?x, ?y) | *is-blocking*(pot, left-door) |
| *is-right-cabinet-door-blocked*(?x) | *is-blocking*(?x, ?y) | *is-blocking*(pot, right-door) |
| - | *is-closed*(?x) | *is-closed*(left-door), *is-closed*(right-door) |
| | | *is-closed*(drawer) |
| - | *is-moved-away*(?x) | *is-moved-away*(pot) |

## A.3 Temporal Segmentation

Before the generation of behavior description, we segment each demonstration into a sequence of *contact-based primitives*. We consider seven primitives describing the interactions between the robot and other objects: *open/close* grippers without holding objects, *move-to*($x$) which moves the gripper to an object, *grasp*($x, y$) and *place*($x, y$) which grasp and place object $x$ from/onto another object $y$, *move*($x$) which moves the currently holding object $x$ and *push*($x$).

We use a set of heuristics to automatically segment the continuous trajectories using proprioception, i.e., gripper open state, and object segmentation. Specifically, *open* and *close* are directly detected by checking whether the gripper width is at the maximum or minimum value. $grasp(x, y)$ and $place(x, y)$ correspond to the other closing and opening gripper actions. $move(x)$, $push(x)$ and $move\text{-}to(x)$ are matched to temporal segments between pairs of gripper actions. Their type can be inferred based on the preceding and following gripper actions. We make a simplifying assumption that the robot moves freely in space only when the gripper is fully open and pushes objects only when the gripper is fully closed. These are given as instructions to the human demonstrators. In the simulator, the arguments of the primitives are obtained from the contact state. In the real world, they are inferred from the language annotations of the actions (e.g.,"place the kettle on the stove" corresponds to *place*(kettle, stove)) procedurally or by the LLMs. The arguments can also be left unspecified; these arguments mainly provide additional contextual information about the target objects.

In Section 4.1, we discuss that we use LLMs to predict a *body* of contact primitive sequence associated with each behavior description. This additional step helps account for noises in the segmentation annotations, which are prevalent in CALVIN's language-annotated demonstrations. For example, the language annotation "lift-block-table" correspond to the contact sequence {*move-to, grasp, move, place*}. Based on the generated *body*, the behavior can be correctly mapped to {*grasp, move*} and the demonstration trajectories can then be re-segmented. This additional step is crucial for learning accurate groundings of the states and actions.

Our approach to temporal segmentation are similar to keyframe-based methods like PerAct [74] and 3D Diffuser Actor [75], which rely on end-effector states (e.g., grasp and release) and velocities to segment demonstrations. However, our method differs by distinguishing between prehensile and non-prehensile manipulation through detecting whether an object is grasped. This enables the development of behaviors such as *move-faucet-away*, where the robot pushes the faucet head without grasping it. In our preliminary studies, we also experiment with other vision-based methods including UVD [76] and Lotus [77]. A main issue for incorporating these methods is that they provide less consistent segmentations for different occurrences of the same behavior. As we discussed in Section 6, more advanced contact detection techniques will be an important future direction for using contact primitives as a meaningful interface between actions and language.

## A.4   Abstract Verification

After the generation of the behavior descriptions, we verify the generated behavior descriptions by performing abstract verification on the demonstration trajectories. Given a segmented sequence of the trajectory where each segment is associated with a behavior, we verify whether the preconditions of each behavior can be satisfied by the accumulated effects of the previous behaviors. Pseudocode for this algorithm is shown in Algorithm 1.

## A.5   Automatic Predicate Annotation

We leverage *all* behavior descriptions to automatically label an observation sequence $\{o_1, ..., o_H\}$ based on its associated segmentation. In particular, at $o_0$, we label all state predicates as "unknown." Next, we unroll the sequence of executed behaviors. As illustrated in Fig. 3c, before applying a behavior $a$ at step $o_t$, we label all predicates in $pre_a$ true and predicates in $eff_a$ false. When $a$ finishes at step $o_{t'}$, we label all predicates in $eff_a$. In addition, we will propagate the labels for state predicates to later time steps until they are explicitly altered by another behavior $a$. Pseudocode for this algorithm is shown in Algorithm 2.

As a result, we obtain both positive and negative examples to train the binary predicate classifiers. In particular, the negative examples of a predicate come from two sources. First, the preconditions and effects of a behavior can include negated predicates. Second, during automatic predicate annotation, we label all predicates in the effects as false.

**Algorithm 1** Abstract Verification

---

**Input:** Dataset $\mathcal{D}$, Behavior descriptions $\mathcal{A}$
 1: *error_counter* ← a counter for sequencing errors related to each behavior
 2: *counter* ← a counter for storing the occurrences of each behavior
 3: **for** $i \leftarrow 1$ to $K$ **do**
 4:     obtain a behavior sequence $\mathcal{D}_i \leftarrow \{a_1^i, ..., a_N^i\}$
 5:     initialize a dictionary for predicate state *pred* ← {}
 6:     **for** $t \leftarrow 1$ to $N$ **do**
 7:         **for** each *exp* in $pre_{a_t^i}$ **do**
 8:             $(p, v) \leftarrow$ EXTRACTPREDICATEANDBOOL($exp$)
 9:             **if** $p$ not in *pred* **then**
10:                 $pred[p] \leftarrow v$
11:             **else**
12:                 **if** $pred[p] \neq v$ **then**
13:                     increment *error_counter*$[a_t^i]$
14:         **for** each *exp* in $eff_{a_t^i}$ **do**
15:             $(p, v) \leftarrow$ EXTRACTPREDICATEANDBOOL($exp$)
16:             $pred[p] \leftarrow v$
17:         increment *counter*$[a_t^i]$
18: **for** each $a$ in *error_counter* **do**
19:     **if** *error_counter*$[a]$/*counter*$[a] >$ *threshold* **then**
20:         regenerate the behavior description for $a$

---

### A.6 Classifier Implementation

Based on the state predicate dataset generated from behavior definitions, we train a set of state classifiers $f_\theta(p) : \mathcal{O} \to \{T, F\}$, which are implemented as standard neural networks for classification.

In the simulation experiment, the classifier model is based on a pre-trained CLIP model (`ViT-B/32`). We use the image pre-processing pipeline from the CLIP model to process the input images. We use images from the static camera in the simulation. We perform one additional step of image processing to mask out the robot arm, which we find in our preliminary experiment to help avoid overfitting. We do not use the global image embedding from the CLIP model, instead we extract the patch tokens from the output of the vision transformer. We downsize the concatenated patch tokens with a multilayer perceptron (MLP) and then concatenate with word embeddings of the predicate arguments (e.g., *red-block*, *table*). The final embedding is then passed through a predicate-specific MLP to output the logit for binary classification. The CLIP model is frozen, while all other learnable parameters are trained.

In the real-world experiment, we find that, with more limited data than simulation, the pre-trained CLIP model often overfits to spurious relations in the training images (e.g., the state of the faucet is entangled with the location of the kettle). We also experiment with a ResNet-50 model pre-trained on ImageNet and find similar behavior. To improve generalization, we choose to focus on relevant objects and regions. We achieve this by using segmented object point clouds. We use open vocabulary object detector Grounding-Dino [68] to detect objects given object names. The predicted 2D bounding boxes are projected into 3D and used to extract regions of the point cloud surrounding each object. The point-cloud-based classifier is based on the shape classification model from the Point Cloud Transformer (PCT) [78]. We concatenate the segmented object point clouds and include one additional channel to indicate the identity of each point. The PCT is used to encode the combined point cloud and output the final logit. The PCT model is trained from scratch.

We also experiment with replacing trained classifiers a general-purpose VLM `gpt-4o`. The VLM fails to robustly determine the boolean values of the predicates in different states. Our observation is that the VLM can more reliably detect the states of articulated objects (e.g., the states of drawers and cabinet doors) than more complex spatial relations (e.g., whether the faucet head is aligned with

---

**Algorithm 2** Predicate Annotation

---

**Input:** Behavior sequence $\{a_1, ..., a_N\}$, Observation sequence $\{o_1, ..., o_H\}$, Descriptions $\mathcal{A}$

1:  *propagated* ← an empty list of propagated predicates
2:  *prev_effs* ← a list for storing effects from previous step
3:  *timed_preds* ← an empty list of predicates associated with time steps
4:  *pred_obs* ← an empty list for storing predicates paired with observations
5:  **for** $t \leftarrow 1$ to $N$ **do**
6:     *// Precondition*
7:     *timed_preds* ← *timed_preds* $\cup$ GETTIMEDPREDICATES($pre_{a_t}, t$)
8:     *timed_preds* ← *timed_preds* $\cup$ GETTIMEDPREDICATES($\neg eff_{a_t}, t$)
9:     *// Propagated*
10:    **for** each $p$ in *propagated* **do**
11:       **if** not ALTERED($p, a_t$) **then**
12:         UPDATETIME($p, t$)
13:       **else**
14:         *propagated.remove*($p$)
15:         *timed_preds.add*($p$)
16:    *// Previous effects*
17:    **for** each $p$ in *prev_effs* **do**
18:       **if** not ALTERED($p, a_t$) **then**
19:         *propagated.add*($p$)
20:       **else**
21:         *timed_preds.add*($p$)
22:    *// Store effects for next step*
23:    *prev_effs* ← GETTIMEDPREDICATES($eff_{a_t}, t$)
24: *timed_preds.update*(*propagated*)
25: *timed_preds.update*(*prev_effs*)
26: **for** each $p$ in *timed_preds* **do**
27:    *pred_obs.update*(MATCHTIMEDPREDICATEWITHOBSERVATION($p, \{o_1, ..., o_H\}$))
28: **return** *pred_obs*

---

the kettle). Following our original experimental procedure, we combine the VLM classifier with our planner to test the generalization cases for the Boil Water domain. Due to the inaccuracy of the state classification, the overall system achieves a 0% success rate.

We imagine future VLMs will become more reliable in detecting the predicate states, and our method will be beneficial to the paradigm of using VLMs as classifiers in the following ways: 1) our automatic predicate annotation method can generate examples for VLMs to recognize new concepts through visual in-context prompting or supervised fine-tuning; 2) our method provides data to train task-specific classifiers based on 3D representations, which are complementary to general-purpose VLMs in recognizing geometric and spatial concepts; 3) our method provides a way to learn user-specific predicates (e.g., a predicate that determines whether clean dishes are arranged according to a user's preferences) from demonstrations. In our preliminary experiment on the Boil Water domain, we find that using examples generated from our method as in-context visual examples improves the VLM's accuracy by around 5% on 220 withheld testing examples.

## A.7 Policy Implementation

For each behavior, we train control policies $\pi_\theta(a) : \mathcal{O} \to \mathcal{U}$, implemented as a diffusion policy [1]. We make three changes to the original implementation to facilitate chaining the learned behaviors. First, when training the model to predict the first raw action for each skill, we replace the history observations with observations sampled randomly from a temporal window prior to when the skill is executed, to avoid bias in the starting positions of the robot arm. Second, we perform biased sampling of the training sequences to ensure that the policy is trained on a diverse set of starting positions. Third, at the end of each training sequence, we append a sequence of zeros actions so the learned

policy can learned to predicate termination. These strategies are implemented for both the simulation and the real world.

In simulation, we construct the point cloud of the scene using the RGB-D image from the frame-mounted camera. We then obtain segmented object point clouds for the relevant objects of each behavior (e.g., *table* and *block* for *pick-block-table*) with groundtruth segmentation masks from the PyBullet simulator. The segmented point clouds of the objects are concatenated to form the input point cloud observation. The model uses the PCT to encode a sequence of point clouds as history observations and uses another time-series transformer encoder to reason over the history observations and predict the next actions. The time-series transformer is similar in design to the transformer-based diffusion policy [1].

In the real world, we use RGB images from four stationary cameras mounted around the workspace and a wrist-mounted camera as input to an image-based diffusion policy model. The input is processed using five separate ResNet-34 encoder heads. The policy directly predicts the gripper pose in the world frame. We found the wrist-mounted camera to be particularly helpful in the real-world setup.

## A.8    Planner Implementation

**Planning over geometric constraints.** Geometric constraints, specifically the collision-free constraints for each action, are handled "in the now," right before an action is executed. This is because in order to classify the geometric constraints, we would need to know the exact pose of all objects in the environments. However, we do not explicitly learn models for predicting the exact location of objects after executing certain behaviors.

Our approach to handle this is to process them in the now. It follows the hierarchical planning strategy [79]. In particular, the precondition for actions is an ordered list. In our case, there are two levels: the second level contains the geometric constraint preconditions and the first level contains the rest of the semantic preconditions. During planning, only the first set of preconditions will be added to the subgoal list. After we have finished planning for the first-level preconditions, we consider the second-level precondition for the first behavior in the resulting plan, by possibly moving other obstacles away.

As an example, let us consider the skill of opening the cabinet door. Its first-level precondition only considers the initial state of the cabinet door (i.e., it should be initially closed). It also has a second-level precondition stating that nothing else should be blocking the door. In the beginning, the planner only considers the first-level preconditions. When this behavior is selected to be executed next, the planner checks for the second-level precondition. If this non-blocking precondition is not satisfied in the current state, we will recursively call the planner to achieve it (which will generate actions that move the blocking obstacles away). If this precondition has already been satisfied, we will proceed to execute the policy associated with this *opening-cabinet-door* skill.

This strategy will work for scenarios where there is enough space for moving obstacles around and the robot does not need to make dedicated plans for arranging objects. In scenarios where space is tight and dedicated object placement planning is required, we can extend our framework to include the prediction of object poses after each skill execution.

**Planning over partial observability.** Partial observability is handled assuming the most likely state. In particular, the effect definitions for all behaviors are deterministic. It denotes the most likely state that it will result in. For example, in the definition of behaviors for finding objects (e.g., the *find-object-in-left-cabinet*), we have a deterministic and "optimistic" effect statement that the object will be visible after executing this action.

At performance time, since we will replan after executing each behavior, if the object is not visible after we have opened the left cabinet, the planner will automatically plan for other actions to achieve this visibility subgoal.

This strategy works for simple partially observable Markov decision processes (POMDPs). A potential extension to it is to model a belief state (e.g., representing a distribution of possible object

poses) and execute belief updates on it. Planners can then use more advanced algorithms such as observation-based planning to generate plans. Such strategies have been studied in task and motion planning literature [79, 80].

# B  Simulation Experiment Details

In the effort to standardize our evaluation, we adopt the standard CALVIN benchmark to evaluate all methods without any modifications to its language annotations, action space, skills, and environments.

## B.1  Task Design

To evaluate generalization to new long-horizon manipulation tasks, we designed six tasks that fall into three categories: Abstract Goal, Geometric Constraint, and Partial Observability. Each task has a language instruction, a sampler that generates random initial states, and a goal satisfaction function for evaluation. We provide details for each task below.

**Task-1**

- **Task Category:** Abstract Goal
- **Language Instruction:** *turn off all lights.*
- **Logical Goal:** (and (is-turned-off led) (is-turned-off lightbulb))
- **Initial State:** Both the led and the lightbulb are initially turned on.
- **Goal Satisfaction:** The logical states of both the lightbulb and the led are off.
- **Variation:** The initial states of the led and the lightbulb are both on and the goal is to turn them off.

**Task-2**

- **Task Category:** Abstract Goal
- **Language Instruction:** *move all blocks to the closed drawer.*
- **Logical Goal:** (and (is-in red-block drawer) (is-in blue-block drawer) (is-in pink-block drawer))
- **Initial State:** The blocks are visible and not in the drawer. The drawer is closed.
- **Goal Satisfaction:** The blocks are in the drawer.

**Task-3**

- **Task Category:** Abstract Goal
- **Language Instruction:** *move all blocks to the open drawer.*
- **Logical Goal:** (and (is-in red-block drawer) (is-in blue-block drawer) (is-in pink-block drawer))
- **Initial State:** The blocks are visible and not in the drawer. The drawer is open.
- **Goal Satisfaction:** The blocks are in the drawer.

**Task-4**

- **Task Category:** Partial Observability
- **Language Instruction:** *place a red block on the table.*
- **Logical Goal:** (is-on red-block table)
- **Initial State:** The red block is in the drawer and the drawer is closed.
- **Goal Satisfaction:** The red block is placed on the table.
- **Variations:** Find the blue block or the pink block.

**Task-5**

- **Task Category:** Partial Observability
- **Language Instruction:** *place a red block on the table.*
- **Logical Goal:** (is-on red-block table)
- **Initial State:** The red block is behind the sliding door.
- **Goal Satisfaction:** The red block is placed on the table.
- **Variations:** Find the blue block or the pink block.

**Task-6**

- **Task Category:** Geometric Constraint
- **Language Instruction:** *open the slider.*
- **Logical Goal:** (is-slider-left slider)
- **Initial State:** The sliding door is on the right and there is a pink block on the path of the sliding door to the left.
- **Goal Satisfaction:** The sliding door is within 5cm of the left end.
- **Variations:** Move the slider to the right.

### B.2 Baseline Implementation

**HULC.** This baseline is a hierarchical policy learning method that learns from language-annotated play data using hindsight labeling [72]. It's one of the best-performing models on the $D \to D$ split of the CALVIN benchmark. We omit the comparison to the HULC++ method [24], the follow-up work of HULC that leverages affordance prediction and motion planning to improve the low-level skills, because our evaluation is focused on the task planning ability of the learned hierarchical model.

**SayCan.** This baseline combines an LLM-based planner that takes the language instruction and learned feasibility functions for skills to perform task planning. SayCan relies on well-trained affordance models to filter out inadmissible actions from the LLMs. Because we learn new skills from demonstrations, it is infeasible to learn Q-functions for these skills through RL. We opt to use two other types of affordance functions*. The first affordance function is based on the detected objects in the scene.. We provide the detected objects in the prompt. The second affordance function is an image-based neural network that predicts the likelihood of taking an action in a physical state, and the network is trained on our demonstration data. The backbone of the neural network is the same as our image-based predicate classifiers. The prompt of the model is listed in Listing 8.

**Robot-VILA.** This baseline performs task planning with a VLM. We adopt the prompts provided in the original paper to the CALVIN environment. The prompts are divided into the initial prompt that is used to generate the task plan given the initial observation (shown in Listing 9) and the follow-up prompt that is used for all subsequent steps (shown in Listing 10). We use `gpt-4-turbo-2024-04-09` as the VLM. Because the model does not memorize the history. We store the history dialogue, including the text input and the image input, and concatenate the history dialogue with the current dialogue as the input to the VLM.

**T2M-Shooting.** This baseline (in particular, the shooting-based algorithm) is similar to the SayCan algorithm except that: 1) it uses a multi-step feasibility model in contrast to the single-step feasibility model used by SayCan; 2) the LLM additionally takes a symbolic state description of object states and relationships. The original Text2Motion method assumes access to ground-truth symbolic states. For comparison, in this paper, we compare Text2Motion with BLADE in two settings: one with the ground-truth states and the other with the state classifiers learned by BLADE. The prompt of the model is listed in Listing 11.

## C   Qualitative Examples

In this section, we include three qualitative examples from the CALVIN experiments to compare the generalization capabilities of BLADE with baselines. Specifically, Fig. A4 shows generalization to abstract goal, Fig. A5 shows generalization to partial observability, and Fig. A6 shows generalization to geometric constraint. In summary, BLADE is able to generate accurate long-horizon manipulation plans for novel situations while latent planning, LLM, and VLM baselines fail.

## D   Real World Experiment Details

We validated our approach in two real-world domains. As shown in Fig. A1, we employ a 7-degree of freedom (DOF) Franka Emika robotic arm equipped with a parallel jaw gripper. A total of Five Intel RealSense RGB-D cameras are used to provide observation for our policies and state classifiers. Four cameras are mounted on the frame and one additional camera is mounted on the robot's wrist.

---

*We closely follow the official open-source SayCan implementation, available at https://github.com/google-research/google-research/tree/master/saycan.

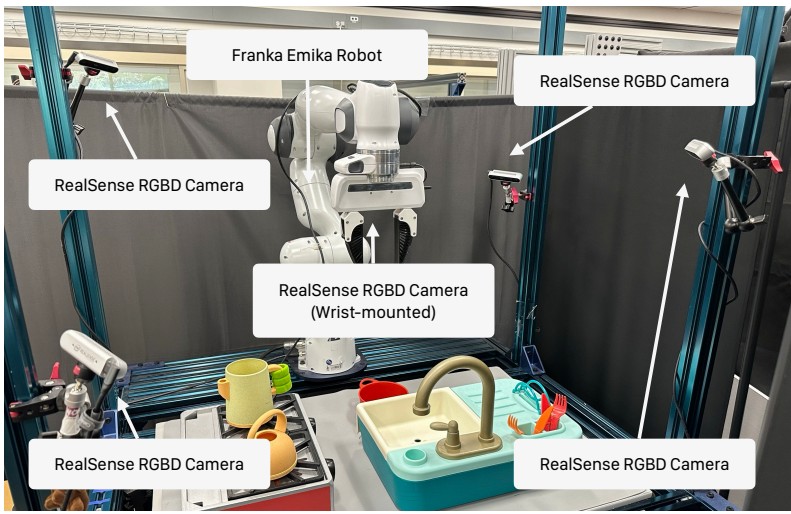

**Figure A1:** We use a 7-degree of freedom (DOF) Franka Emika robotic arm with a parallel jaw gripper for our real-world experiment. A total of Five Intel RealSense RGB-D cameras are used to provide observation for our policies and state classifiers. Four cameras are mounted on the frame and an additional one is mounted to the robot's wrist.

Our teleoperation system uses a 3DConnexion SpaceMouse for control. During the collection of demonstrations, we record the the pose of the end effector, the gripper width, and the RGB-D images from the five cameras. We collected 85 demonstrations for each of the two real-world domains, which provide the training data for the diffusion policy models and the state classifiers.

### D.1 Task Design

Similar to our simulation experiments, our evaluation protocol includes the design of six tasks aimed at assessing the model's generalization capabilities across new long-horizon tasks. These tasks are specifically crafted to test the model's proficiency for four types of generalization: Unseen Initial Condition, State Perturbation, Partial Observability, and Geometric Constraint.

**Task-1**

- **Domain:** Boil Water
- **Task Category:** Unseen Initial Condition
- **Language Instruction:** *Fill the kettle with water and place it on the stove*
- **Logical Goal:** (and (is-filled kettle) (is-placed-on kettle stove) (is-turned-off faucet-knob))
- **Initial State:** The kettle is placed inside the sink, and the stove is not blocked. The faucet is turned off with the faucet head turned away.

**Task-2**

- **Domain:** Boil Water
- **Task Category:** State Perturbation
- **Language Instruction:** *Fill the kettle with water and place it on the stove*
- **Logical Goal:** (and (is-filled kettle) (is-placed-on kettle stove) (is-turned-off faucet-knob))
- **Initial State:** The kettle is placed inside the sink and the stove is blocked.
- **Perturbation**: The human user moves the kettle from the sink to the table after the robot turns the faucet head towards the sink. The robot needs to replan to move the kettle back to the sink.

**Task-3**

- **Domain:** Boil Water
- **Task Category:** Geometric Constraint
- **Language Instruction:** *Fill the kettle with water and place it on the stove*
- **Logical Goal:** (and (is-filled kettle) (is-placed-on kettle stove) (is-turned-off faucet-knob))

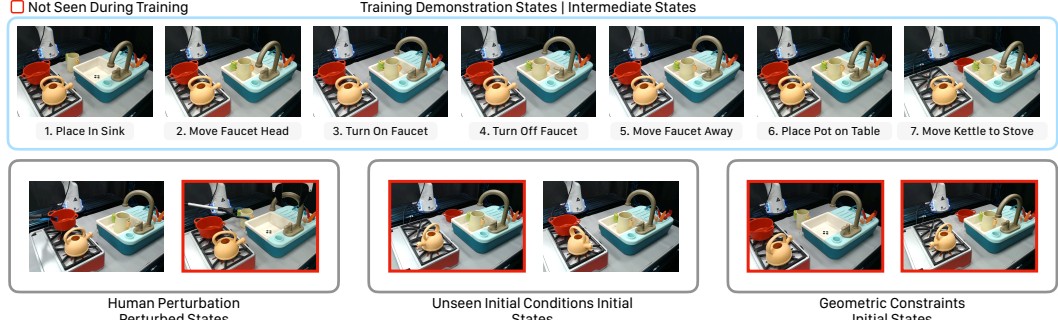

**Figure A2:** Training and testing states for the Boil Water domain.

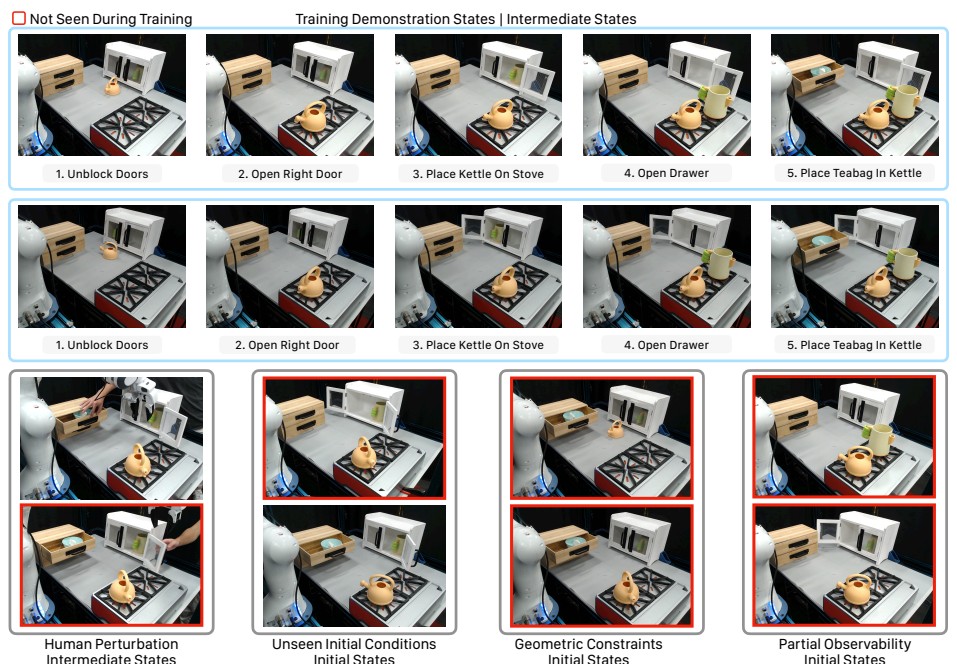

**Figure A3:** Training and testing states for the Make Tea domain.

- **Initial State:** The kettle is placed inside the sink and the stove is blocked, creating a geometric constraint.

**Task-4**

- **Domain:** Make Tea
- **Task Category:** Unseen Initial Condition
- **Language Instruction:** *Place the kettle on the stove and place the teabag inside the kettle.*
- **Logical Goal:** (and (is-placed-on kettle stove) (is-placed-inside teabag kettle))
- **Initial State:** The kettle is placed inside a cabinet. The cabinet doors are open. The drawer is closed.

**Task-5**

- **Domain:** Make Tea
- **Task Category:** State Perturbation
- **Language Instruction:** *Place the kettle on the stove and place the teabag inside the kettle.*
- **Logical Goal:** (and (is-placed-on kettle stove) (is-placed-inside teabag kettle))
- **Initial State:** The kettle is placed inside the cabinet and the cabinet door is open. The drawer is initially closed.
- **Perturbation**: Once the robot opens the drawer, a human user closes the drawer.

**Task-6**

- **Domain:** Make Tea
- **Task Category:** Geometric Constraint
- **Language Instruction:** *Place the kettle on the stove and place the teabag inside the kettle.*
- **Logical Goal:** (and (is-placed-on kettle stove) (is-placed-inside teabag kettle))
- **Initial State:** There is a teapot blocking the cabinet doors. The kettle is inside the cabinet. The drawer is open with the teabag visible.

**Task-7**

- **Domain:** Make Tea
- **Task Category:** Partial Observability
- **Language Instruction:** *Place the kettle on the stove and place the teabag inside the kettle.*
- **Logical Goal:** (and (is-placed-on kettle stove) (is-placed-inside teabag kettle))
- **Initial State:** The kettle is placed inside a cabinet and is not visible.

### D.2 Qualitative Examples of Novel States

In Fig. A2 and Fig. A3, we visualize and confirm that more than half of the initial states and the perturbed states are not part of the demonstrations in our experiments; therefore, purely imitation-learning-based methods will struggle to solve.

## E Prompts for Baselines

In this section, we provide the prompts for the baselines used in the simulation experiments. We provide the prompts for SayCan in Listing 8, Robot-VILA in Listing 9 and Listing 10, and T2M-Shooting in Listing 11.

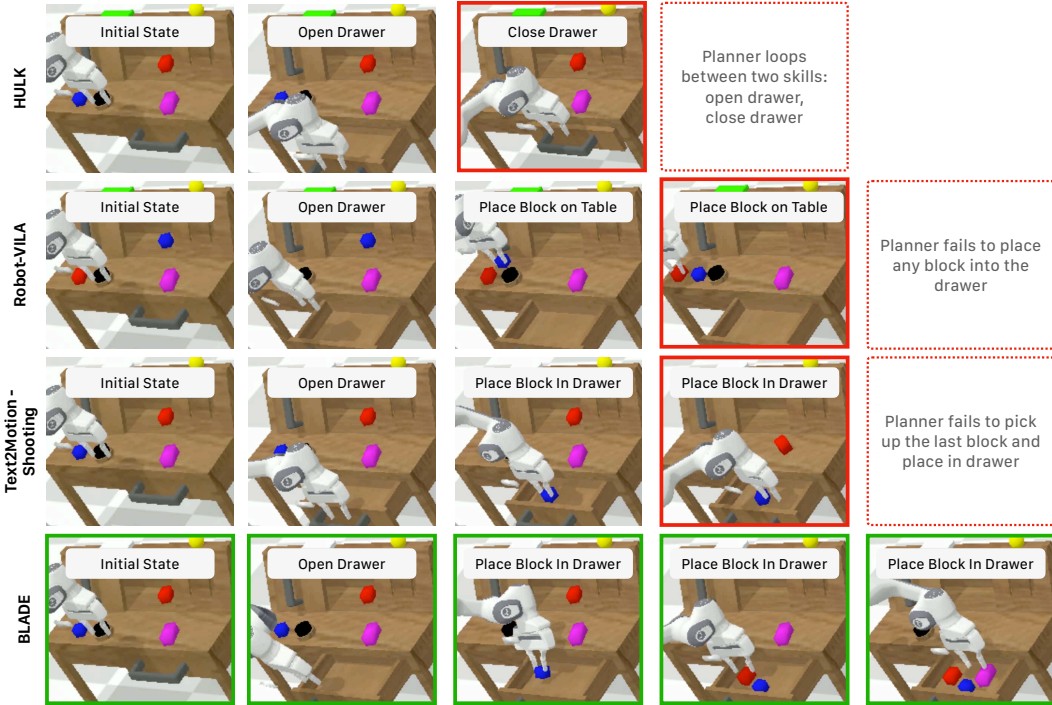

**Figure A4:** BLADE and baseline performance on an Abstract Goal generalization task in the CALVIN environment.

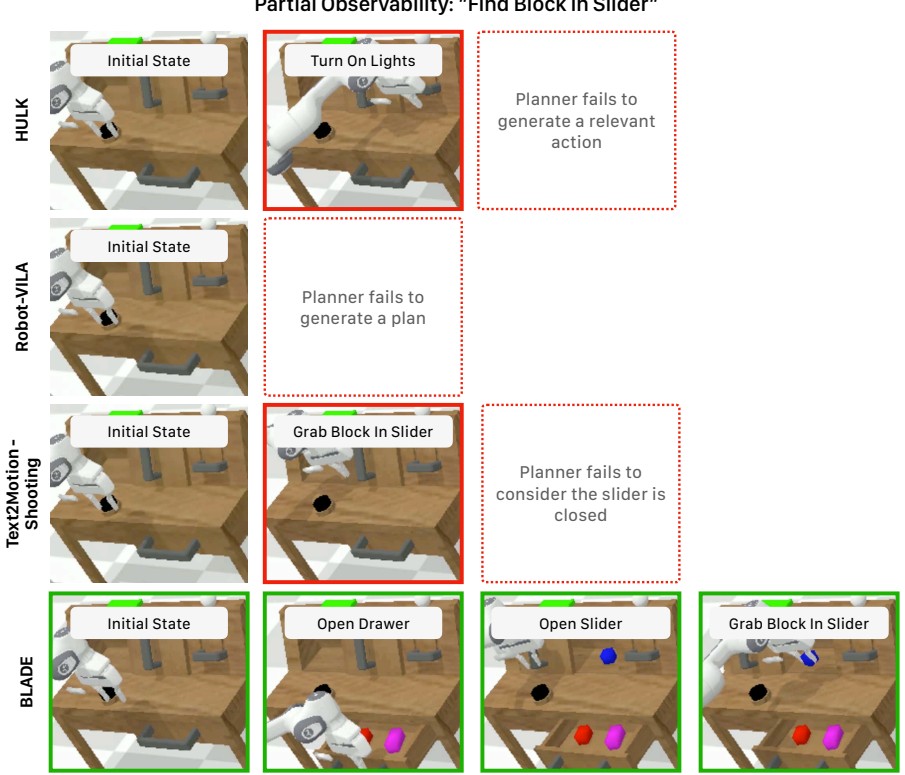

**Figure A5:** BLADE and baseline performance on the Partial Observability generalization task in the CALVIN environment.

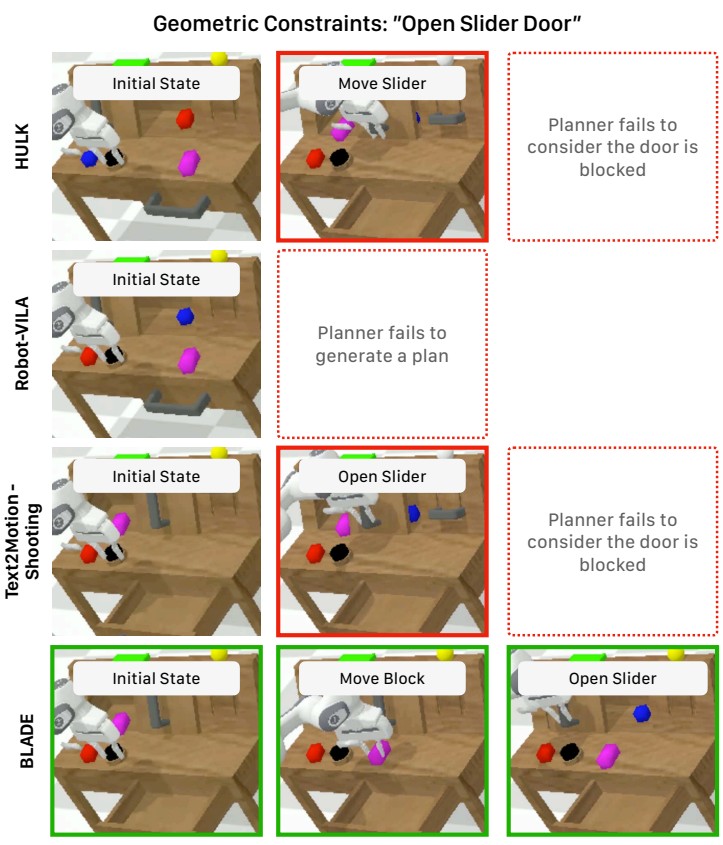

**Figure A6:** BLADE and baseline performance on the Geometric Constraint generalization task in the CALVIN environment.

**Listing 1: Behavior descriptions generated by the LLM for the CALVIN domain.**

```
;; lift_block_table
(:action lift-block-table
 :parameters (?block - item ?table - item)
 :precondition (and (is-block ?block) (is-table ?table) (is-on ?block ?table) (not (is-lifted
 ?block)))
 :effect (and (lifted ?block) (not (is-on ?block ?table)))
 :body (then
   (grasp ?block ?table)
   (move ?block)
 )
)

;; lift_block_slider
(:action lift_block_slider
 :parameters (?block - item ?slider - item)
 :precondition (and (is-block ?block) (is-slider ?slider) (is-in ?block ?slider))
 :effect (and (lifted ?block) (not (is-in ?block ?slider)))
 :body (then
   (grasp ?block ?slider)
   (move ?block)
 )
)

;; lift_block_drawer
(:action lift-block-drawer
 :parameters (?block - item ?drawer - item)
 :precondition (and (is-block ?block) (is-drawer ?drawer) (is-in ?block ?drawer) (is-open ?
 drawer))
 :effect (and (lifted ?block) (not (is-in ?block ?drawer)))
 :body (then
   (grasp ?block ?drawer)
   (move ?block)
 )
)

;; place_in_slider
(:action place-in-slider
 :parameters (?block - item ?slider - item)
 :precondition (and (is-block ?block) (is-slider ?slider) (is-lifted ?block))
 :effect (and (is-in ?block ?slider) (not (is-lifted ?block)))
 :body (then
   (place ?block ?slider)
 )
)

;; place_in_drawer
(:action place-in-drawer
 :parameters (?block - item ?drawer - item)
 :precondition (and (is-block ?block) (is-drawer ?drawer) (is-lifted ?block) (is-open ?drawer)
 )
 :effect (and (is-in ?block ?drawer) (not (is-lifted ?block)))
 :body (then
   (place ?block ?drawer)
 )
)

;; place_on_table
(:action place-on-table
 :parameters (?block - item ?table - item)
 :precondition (and (is-block ?block) (is-table ?table) (is-lifted ?block))
 :effect (and (is-on ?block ?table) (not (is-lifted ?block)))
 :body (then
   (place ?block ?table)
 )
)

;; stack_block
(:action stack_block
 :parameters (?block - item ?target - item)
 :precondition (and (is-block ?block) (is-block ?target) (is-lifted ?block))
 :effect (and (stacked ?block ?target) (not (is-lifted ?block)))
 :body (then
   (place ?block ?target)
 )
)

;; unstack_block
(:action unstack_block
```

```
 :parameters (?block1 - item ?block2 - item)
 :precondition (and (is-block ?block1) (is-block ?block2) (stacked ?block1 ?block2))
 :effect (and (unstacked ?block1 ?block2) (is-lifted ?block1) (not (stacked ?block1 ?block2)))
 :body (then
   (grasp ?block1 ?block2)
   (move ?block1)
 )
)

;; rotate_block_right
(:action rotate-block-right
 :parameters (?block - item ?table - item)
 :precondition (and (is-block ?block) (is-table ?table) (is-on ?block ?table))
 :effect (and
          (rotated-right ?block)
          (not (rotated-left ?block)))
 :body (then
   (grasp ?block ?table)
   (move ?block)
   (place ?block ?table)
 )
)

;; rotate_block_left
(:action rotate_block_left
 :parameters (?block - item ?table - item)
 :precondition (and (is-block ?block) (is-table ?table) (is-on ?block ?table))
 :effect (and (rotated-left ?block))
 :body (then
   (grasp ?block)
   (move ?block)
   (place ?block)
 )
)

;; push_block_right
(:action push_block_right
 :parameters (?block - item ?table - item)
 :precondition (and (is-block ?block) (is-table ?table) (is-on ?block ?table))
 :effect (and (pushed-right ?block) (not (pushed-left ?block)))
 :body (then
   (close)
   (push ?block)
   (open)
 )
)

;; push_block_left
(:action push-block-left
 :parameters (?block - item)
 :precondition (and (is-block ?block))
 :effect (and (pushed-left ?block))
 :body (then
   (close)
   (push ?block)
   (open)
 )
)

;; move_slider_left
(:action move_slider_left
 :parameters (?slider - item)
 :precondition (and (is-slider ?slider) (is-slider-right ?slider))
 :effect (and (is-slider-left ?slider) (not (is-slider-right ?slider)))
 :body (then
   (grasp ?slider)
   (move ?slider)
   (place ?slider)
 )
)

;; move_slider_right
(:action move-slider-right
 :parameters (?slider - item)
 :precondition (and (is-slider ?slider) (not (is-slider-right ?slider)))
 :effect (and (is-slider-right ?slider))
 :body (then
   (grasp ?slider)
   (move ?slider)
   (place ?slider)
 )
```

```
)

;; open_drawer
(:action open-drawer
 :parameters (?drawer - item)
 :precondition (and (is-drawer ?drawer) (is-close ?drawer))
 :effect (and (is-open ?drawer) (not (is-close ?drawer)))
 :body (then
   (close)
   (push ?drawer)
   (open)
 )
)

;; close_drawer
(:action close-drawer
 :parameters (?drawer - item)
 :precondition (and (is-drawer ?drawer) (is-open ?drawer))
 :effect (and (is-close ?drawer) (not (is-open ?drawer)))
 :body (then
   (close)
   (push ?drawer)
   (open)
 )
)

;; turn_on_lightbulb
(:action turn-on-lightbulb
 :parameters (?lightbulb - item)
 :precondition (and (is-lightbulb ?lightbulb) (is-turned-off ?lightbulb))
 :effect (and (is-turned-on ?lightbulb) (not (is-turned-off ?lightbulb)))
 :body (then
   (close)
   (push ?lightbulb)
   (open)
 )
)

;; turn_off_lightbulb
(:action turn-off-lightbulb
 :parameters (?lightbulb - item)
 :precondition (and (is-lightbulb ?lightbulb) (is-turned-on ?lightbulb))
 :effect (and (is-turned-off ?lightbulb) (not (is-turned-on ?lightbulb)))
 :body (then
   (close) (push ?lightbulb) (open)
 )
)

;; turn_on_led
(:action turn-on-led
 :parameters (?led - item)
 :precondition (is-led ?led)
 :effect (and (is-turned-on ?led) (not (is-turned-off ?led)))
 :body (then
   (close)
   (push ?led)
   (open)
 )
)

;; turn_off_led
(:action turn-off-led
 :parameters (?led - item)
 :precondition (and (is-led ?led) (is-turned-on ?led))
 :effect (and (is-turned-off ?led) (not (is-turned-on ?led)))
 :body (then
   (close)
   (push ?led)
   (open)
 )
)

;; push_into_drawer
(:action push-into-drawer
 :parameters (?block - item ?drawer - item)
 :precondition (and (is-block ?block) (is-drawer ?drawer) (is-open ?drawer))
 :effect (and (is-in ?block ?drawer))
 :body (then
   (close)
   (push ?block)
   (open)
```

```
  )
)
```

---

---

```
**Primitive Actions:**
There are seven primitive actions that the robot can perform. They are:
- (grasp ?x ?y): ?x and ?y are two object variables. ?x is the object that the robot will be
grasping, ?y is the object that ?x is currently on or in.
- (place ?x ?y): ?x and ?y are two object variables. ?x is the object that the robot is
currently holding, ?y is the object that ?x will be placed on or in.
- (move ?x): ?x is the object that the robot is currently holding and will be moved by the
robot.
- (push ?x): ?x is the object that the robot will be pushing.
- (move-to ?x): the robot arm will move without holding any object or pushing any object.
- (open): the robot gripper will open fully.
- (close): the robot gripper will close without grasping any object.

**Combined Primitives:**
The primitive actions can be combined into a high-level routine. For example, (then (grasp ?x
?y) (move ?x) (place ?x ?y)) means the robot will pick up ?x from ?y, move ?x, and place ?x to
 ?z. The possible combination of primitives are:
A. (then (grasp ?x ?y) (move ?x))
B. (then (place ?x ?y))
C. (then (grasp ?x ?y) (move ?x) (place ?x ?z))
D. (then (close) (push ?x) (open))
```

---

---

```
**Predicates for Preconditions and Effects:**
The list of all possible predicates for defining the preconditions and effects of the high-
level routine are listed below:

For specifying the type of the object:
- (is-table ?x - item): ?x is a table.
- (is-slider ?x - item): ?x is a slider.
- (is-drawer ?x - item): ?x is a drawer.
- (is-lightbulb ?x - item): ?x is a lightbulb.
- (is-led ?x - item): ?x is a led.
- (is-block ?x - item): ?x is a block.

For specifying the attributes of the object:
- (is-red ?x - item): ?x is red. This predicate applies to a block.
- (is-blue ?x - item): ?x is blue. This predicate applies to a block.
- (is-pink ?x - item): ?x is pink. This predicate applies to a block.

For specifying the state of the object:
- (rotated-left ?x - item): ?x is rotated left. This predicate applies to a block.
- (rotated-right ?x - item): ?x is rotated right. This predicate applies to a block.
- (pushed-left ?x - item): ?x is pushed left. This predicate applies to a block.
- (pushed-right ?x - item): ?x is pushed right. This predicate applies to a block.
- (lifted ?x - item): ?x is lifted in the air. This predicate applies to a block.
- (is-open ?x - item): ?x is open. This predicate applies to a drawer.
- (is-close ?x - item): ?x is close. This predicate applies to a drawer.
- (is-turned-on ?x - item): ?x is turned on. This predicate applies to a lightbulb or a led.
- (is-turned-off ?x - item): ?x is turned off. This predicate applies to a lightbulb or a led.
- (is-slider-left ?x - item): the sliding door of the slider ?x is on the left.
- (is-slider-right ?x - item): the sliding door of the slider ?x is on the right.

For specifying the relationship between objects:
- (is-on ?x - item ?y - item): ?x is on top of ?y. This predicate applies when ?x is a block
and ?y is a table.
- (is-in ?x - item ?y - item): ?x is inside of ?y. This predicate applies when ?x is a block
and ?y is a drawer or a slider.
- (stacked ?x - item ?y - item): ?x is stacked on top of ?y. This predicate applies when ?x
and ?y are blocks.
- (unstacked ?x - item ?y - item): ?x is unstacked from ?y. This predicate applies when ?x and
 ?y are blocks.

**Task Environment:**
In the environment where the demonstrations are being performed, there are the following
objects:
- A table. Objects can be placed on the table.
- A drawer that can be opened. Objects can be placed into the drawer when it is open.
- A slider which is a cabinet with a sliding door. The sliding door can be moved to the left
or to the right. Objects can be placed into the slider no matter the position of the sliding
door.
- A lightbulb that be can turned on/off with a button.
- A led that can be turned on/off with a button.
```

- Three blocks that can be rotated, pushed, lifted, and placed.

---

**Listing 4: Example Prompt for CALVIN–In-Context Example.**

---

```
**Demonstration Parsing:**
Now, you will help to parse several human demonstrations of the robot performing a task and
generate a lifted description of how to accomplish this task.
For each demonstration, a sequence of performed primitives will be given, with actual object
names. Three demonstrations for the task of "place_in_slider" is:

<code name="primitive_sequence">
primitives = [
  {"name": "grasp", "arguments": ["red_block", "table"]}
  {"name": "move", "arguments": ["red_block"]}
  {"name": "place", "arguments": ["red_block", "slider"]}
  {"name": "move-to", "arguments": [""]}
]


<code name="primitive_sequence">
primitives = [
  {"name": "grasp", "arguments": ["blue_block", "table"]}
  {"name": "move", "arguments": ["blue_block"]}
  {"name": "place", "arguments": ["blue_block", "slider"]}
  {"name": "move-to", "arguments": [""]}
]


<code name="primitive_sequence">
primitives = [
  {"name": "grasp", "arguments": ["pink_block", "table"]}
  {"name": "move", "arguments": ["pink_block"]}
  {"name": "place", "arguments": ["pink_block", "slider"]}
  {"name": "move-to", "arguments": [""]}
]


**Previous Tasks:**
A list of tasks that can be performed before the current task will also be provided as context
. For the task of "place_in_slider", the possible previous tasks are:
lift_block_table, lift_block_drawer, move_slider_right

**Example Output:**
You should generate a lifted description, treating all objects as variables. For example, the
lifted description for "place_in_slider" is:
<code name="mechanism">
(:mechanism place-in-slider
 :parameters (?block - item ?slider - item)
 :precondition (and (is-block ?block) (is-slider ?slider) (is-lifted ?block))
 :effect (and (is-in ?block ?slider) (not (is-lifted ?block)))
 :body (then
   (place ?block ?slider)
 )
)

```

---

**Listing 5: Example Prompt for CALVIN–Instructions.**

---

```
**Think Step-by-Step:**
To generate the lifted description, you should think through the task in natural language in
the following steps. Be EXTREMELY CAREFUL to think through step 3a, 3b, and 4a, 4b.
1. Parse the goal. For example "place_in_slider", the goal is to place a block into the slider
.
2. Think about the possible effects achieved by previous tasks and the previous actions that
have been performed. For "lift_block_table", a block is lifted from the table and the effect
is that the block is lifted. For "lift_block_drawer", a block is lifted from the drawer and
the effect is that the block is lifted. For "move_slider_right", the sliding door of the
slider is moved to the right and the effect is that the sliding door is on the right.
3. Parse the demonstrations and choose the combination of primitives for the current task. The
 demonstrations are noisy so that the demonstrated primitive sequences may include extra
primitive actions that are not necessary for the current task at the beginning or end. The
extra primitive actions can be for the previous tasks. Combining with the understanding of the
 task and previous task to infer the correct combination of primitives for the current task.
3a. In this case, the previous tasks are relevant to the current task. We should think about
how to sequence the previous tasks with the current task. The primitive combination for the
current task should not include primitive actions that have been performed. The above example
 for "place_in_slider" is this case. We can infer that "grasp" in the demonstrated sequences is
 likely to be for the previous tasks and should not be included in the primitive combination
```

for the current task. We therefore choose B. (then (place ?x ?y)). The semantics is that the
robot place the lifted block in the slider.
3b. In this case, the previous tasks are not relevant to the current task.
4. Think about the preconditions. Also specify the types of all relevant objects in the
preconditions.
4a. In this case, previous tasks are relevant to the current task. We should think about the
effects of the previous tasks. For "place_in_slider", the effects of previous tasks include
the block is already lifted. So we should specify that the block is lifted in the
preconditions for the current task.
4b. In this case, previous tasks are not relevant to the current task.
5. Think about the effects. For "place_in_slider", the effects are that the block is in the
slider and the block is not lifted.
6. Write down the mechanism in the format of the example.

**Additional Instructions:**
1. Make sure the generated lifted description starts with <code name="mechanism"> and ends
with .
2. Please do not invent any new predicates for the precondition and effect. You can only use
the predicates listed above.
3. Consider the physical constraints of the objects. For example, a robot arm can not go
through a closed door.
4. For each parameter in :parameters, you should use one of the predicates for specifying the
type of the object to indicate its type (e.g., is-drawer, is-block, and etc).

---

## Listing 6: Example Prompt for CALVIN–Task Input.

```
**Current Task:** place_in_drawer

**Example Sequences:**
<code name="primitive_sequence">
primitives = [
  {"name": "grasp", "arguments": ["blue_block", "table"]}
  {"name": "move", "arguments": ["blue_block"]}
  {"name": "place", "arguments": ["blue_block", "drawer"]}
  {"name": "move-to", "arguments": [""]}
]


<code name="primitive_sequence">
primitives = [
  {"name": "grasp", "arguments": ["red_block", "table"]}
  {"name": "move", "arguments": ["red_block"]}
  {"name": "place", "arguments": ["red_block", "drawer"]}
  {"name": "move-to", "arguments": [""]}
]


<code name="primitive_sequence">
primitives = [
  {"name": "grasp", "arguments": ["pink_block", "table"]}
  {"name": "move", "arguments": ["pink_block"]}
  {"name": "place", "arguments": ["pink_block", "drawer"]}
  {"name": "move-to", "arguments": [""]}
]


**Previous Tasks:** push_into_drawer, lift_block_table, lift_block_slider
```

---

## Listing 7: Example Prompt for Predicate Generation.

```
You are a helpful agent in helping a robot interpret human demonstrations and discover a
generalized high-level routine to accomplish a given task.
**Primitive Actions:**
There are seven primitive actions that the robot can perform. They are:
- (grasp ?x ?y): ?x and ?y are two object variables. ?x is the object that the robot will be
grasping, ?y is the object that ?x is currently on or in.
- (place ?x ?y): ?x and ?y are two object variables. ?x is the object that the robot is
currently holding, ?y is the object that ?x will be placed on or in.
- (move ?x): ?x is the object that the robot is currently holding and will be moved by the
robot.
- (push ?x): ?x is the object that the robot will be pushing.
- (move-to ?x): the robot arm will move without holding any object or pushing any object.
- (open): the robot gripper will open fully.
- (close): the robot gripper will close without grasping any object.

**Task Environment:**
In the environment where the demonstrations are being performed, there are the following
objects:
- A table. Objects can be placed on the table.
```

- A drawer that can be opened. Objects can be placed into the drawer when it is open.
- A slider which is a cabinet with a sliding door. The sliding door can be moved to the left
or to the right. Objects can be placed into the slider no matter the position of the sliding
door.
- A lightbulb that be can turned on/off with a button.
- A led that can be turned on/off with a button.
- Three blocks that can be rotated, pushed, lifted, and placed.

**Task**
You will help the robot to write PDDL definitions for the following actions:
1. lift_red_block_table
2. lift_red_block_slider
3. lift_red_block_drawer
4. lift_blue_block_table
5. lift_blue_block_slider
6. lift_blue_block_drawer
7. lift_pink_block_table
8. lift_pink_block_slider
9. lift_pink_block_drawer
10. stack_block
11. unstack_block
12. place_in_slider
13. place_in_drawer
14. place_on_table
15. rotate_red_block_right
16. rotate_red_block_left
17. rotate_blue_block_right
18. rotate_blue_block_left
19. rotate_pink_block_right
20. rotate_pink_block_left
21. push_red_block_right
22. push_red_block_left
23. push_blue_block_right
24. push_blue_block_left
25. push_pink_block_right
26. push_pink_block_left
27. move_slider_left
28. move_slider_right
29. open_drawer
30. close_drawer
31. turn_on_lightbulb
32. turn_off_lightbulb
33. turn_on_led
34. turn_off_led

Before writing the operators, define the predicates that should be used to write the
preconditions and effects of the operators. Group the predicates into unary predicates that
define the states of objects and binary relations that specify relations between two objects.
For each predicate, list actions that are relevant.

---

## Listing 8: Prompt for SayCan.

---

**Objective:**
You are a helpful agent in helping a robot plan a sequence of actions to accomplish a given
task.
I will first provide context and then provide an example of how to perform the task.

**Task Environment:**
In the robot's environment, there are the following objects:
- A table. Objects can be placed on the table.
- A drawer that can be opened. Objects can be placed into the drawer when it is open.
- A slider which is a cabinet with a sliding door. The sliding door can be moved to the left
or to the right. Objects can be placed into the slider no matter the position of the sliding
door.
- A lightbulb that be can turned on/off with a button.
- A led that can be turned on/off with a button.
- Three blocks that can be rotated, pushed, lifted, and placed.

**Actions:**
There are the following actions that the robot can perform. They are:
- lift_red_block_table: lift the red block from the table.
- lift_red_block_slider: lift the red block from the slider.
- lift_red_block_drawer: lift the red block from the drawer.
- lift_blue_block_table: lift the blue block from the table.
- lift_blue_block_slider: lift the blue block from the slider.
- lift_blue_block_drawer: lift the blue block from the drawer.
- lift_pink_block_table: lift the pink block from the table.
- lift_pink_block_slider: lift the pink block from the slider.
- lift_pink_block_drawer: lift the pink block from the drawer.
- stack_block: stack the blocks.

```
- place_in_slider: place the block in the slider.
- place_in_drawer: place the block in the drawer.
- place_on_table: place the block on the table.
- rotate_red_block_right: rotate the red block to the right.
- rotate_red_block_left: rotate the red block to the left.
- rotate_blue_block_right: rotate the blue block to the right.
- rotate_blue_block_left: rotate the blue block to the left.
- rotate_pink_block_right: rotate the pink block to the right.
- rotate_pink_block_left: rotate the pink block to the left.
- push_red_block_right: push the red block to the right.
- push_red_block_left: push the red block to the left.
- push_blue_block_right: push the blue block to the right.
- push_blue_block_left: push the blue block to the left.
- push_pink_block_right: push the pink block to the right.
- push_pink_block_left: push the pink block to the left.
- move_slider_left: move the slider to the left.
- move_slider_right: move the slider to the right.
- open_drawer: open the drawer.
- close_drawer: close the drawer.
- turn_on_lightbulb: turn on the lightbulb.
- turn_off_lightbulb: turn off the lightbulb.
- turn_on_led: turn on the led.
- turn_off_led: turn off the led.
- do_nothing: do nothing.

**Example Task:**
Now, you will help to parse the goal predicate and generate a list of candidate actions the
robot can potentially take to accomplish the task. You should rank the actions in terms of how
 likely they are to be performed next.
Goal predicate: (is-turned-off led)
Task output:
```python
['turn_off_led', 'do_nothing']
```
In this example above, if the led is on, the robot should turn it off. If the led is already
off, the robot should do nothing.

When the robot successfully completes an action, the robot will ask for the next action to
take.
Considering the executed task: turn_off_led
Your output should be:
```python
['do_nothing']
```
Since the led is already off, the robot should do nothing.

**Additional Instructions:**
1. Make sure the generated plan is a list of actions. Place the list between ```python and
ends with ```.
2. Think Step-by-Step.

Goal predicate: {Based on the given task}
Current symbolic state: {Based on the simulator state}
Executed actions: {Based on the previously executed actions}
```

**Listing 9: Initial Prompt for Robot-VILA.**

```
You are highly skilled in robotic task planning, breaking down intricate and long-term tasks
into distinct primitive actions.
If the object is in sight, you need to directly manipulate it. If the object is not in sight,
you need to use primitive skills to find the object
first. If the target object is blocked by other objects, you need to remove all the blocking
objects before picking up the target object. At
the same time, you need to ignore distracters that are not related to the task. And remember
your last step plan needs to be "done".

Consider the following skills a robotic arm can perform.
- lift_red_block_table: lift the red block from the table.
- lift_red_block_slider: lift the red block from the slider.
- lift_red_block_drawer: lift the red block from the drawer.
- lift_blue_block_table: lift the blue block from the table.
- lift_blue_block_slider: lift the blue block from the slider.
- lift_blue_block_drawer: lift the blue block from the drawer.
- lift_pink_block_table: lift the pink block from the table.
- lift_pink_block_slider: lift the pink block from the slider.
- lift_pink_block_drawer: lift the pink block from the drawer.
- stack_block: stack the blocks.
- place_in_slider: place the block in the slider.
- place_in_drawer: place the block in the drawer.
- place_on_table: place the block on the table.
```

```
- rotate_red_block_right: rotate the red block to the right.
- rotate_red_block_left: rotate the red block to the left.
- rotate_blue_block_right: rotate the blue block to the right.
- rotate_blue_block_left: rotate the blue block to the left.
- rotate_pink_block_right: rotate the pink block to the right.
- rotate_pink_block_left: rotate the pink block to the left.
- push_red_block_right: push the red block to the right.
- push_red_block_left: push the red block to the left.
- push_blue_block_right: push the blue block to the right.
- push_blue_block_left: push the blue block to the left.
- push_pink_block_right: push the pink block to the right.
- push_pink_block_left: push the pink block to the left.
- move_slider_left: move the slider to the left.
- move_slider_right: move the slider to the right.
- open_drawer: open the drawer.
- close_drawer: close the drawer.
- turn_on_lightbulb: turn on the lightbulb.
- turn_off_lightbulb: turn off the lightbulb.
- turn_on_led: turn on the led.
- turn_off_led: turn off the led.
- done: the goal has reached.

You are only allowed to use the provided skills. You can first itemize the task-related
objects to help you plan.
For the actions you choose, list them as a list in the following format.


['turn_off_led', 'open_drawer', 'done']

```

---

**Listing 10: Follow-Up Prompt for Robot-VILA.**

---

```
This image displays a scenario after you have executed some steps from the plan generated
earlier. When interacting with people, sometimes the robotic arm needs to wait for the person'
s action. If you do not find the target object in the current image, you need to continue
searching elsewhere. Continue to generate the plan given the updated environment state.
```

---

**Listing 11: Prompt for Text2Motion.**

---

```
**Objective:**
You are a helpful agent in helping a robot plan a sequence of actions to accomplish a given
task.
I will first provide context and then provide an example of how to perform the task.

**Task Environment:**
In the robot's environment, there are the following objects:
- A table. Objects can be placed on the table.
- A drawer that can be opened. Objects can be placed into the drawer when it is open.
- A slider which is a cabinet with a sliding door. The sliding door can be moved to the left
 or to the right. Objects can be placed into the slider no matter the position of the
sliding door.
- A lightbulb that be can turned on/off with a button.
- A led that can be turned on/off with a button.
- Three blocks that can be rotated, pushed, lifted, and placed.

**Predicates for symbolic state:**
The list of all possible predicates for defining the symbolic state are listed below:
- (rotated-left ?x - item): ?x is rotated left. This predicate applies to a block.
- (rotated-right ?x - item): ?x is rotated right. This predicate applies to a block.
- (pushed-left ?x - item): ?x is pushed left. This predicate applies to a block.
- (pushed-right ?x - item): ?x is pushed right. This predicate applies to a block.
- (lifted ?x - item): ?x is lifted in the air. This predicate applies to a block.
- (is-open ?x - item): ?x is open. This predicate applies to a drawer.
- (is-close ?x - item): ?x is close. This predicate applies to a drawer.
- (is-turned-on ?x - item): ?x is turned on. This predicate applies to a lightbulb or a led.
- (is-turned-off ?x - item): ?x is turned off. This predicate applies to a lightbulb or a
led.
- (is-slider-left ?x - item): the sliding door of the slider ?x is on the left.
- (is-slider-right ?x - item): the sliding door of the slider ?x is on the right.
- (is-on ?x - item ?y - item): ?x is on top of ?y. This predicate applies when ?x is a block
 and ?y is a table.
- (is-in ?x - item ?y - item): ?x is inside of ?y. This predicate applies when ?x is a block
 and ?y is a drawer or a slider.
- (stacked ?x - item ?y - item): ?x is stacked on top of ?y. This predicate applies when ?x
and ?y are blocks.
- (unstacked ?x - item ?y - item): ?x is unstacked from ?y. This predicate applies when ?x
and ?y are blocks.

**Actions:**
```

There are the following actions that the robot can perform. They are:
- lift_red_block_table: lift the red block from the table.
- lift_red_block_slider: lift the red block from the slider.
- lift_red_block_drawer: lift the red block from the drawer.
- lift_blue_block_table: lift the blue block from the table.
- lift_blue_block_slider: lift the blue block from the slider.
- lift_blue_block_drawer: lift the blue block from the drawer.
- lift_pink_block_table: lift the pink block from the table.
- lift_pink_block_slider: lift the pink block from the slider.
- lift_pink_block_drawer: lift the pink block from the drawer.
- stack_block: stack the blocks.
- place_in_slider: place the block in the slider.
- place_in_drawer: place the block in the drawer.
- place_on_table: place the block on the table.
- rotate_red_block_right: rotate the red block to the right.
- rotate_red_block_left: rotate the red block to the left.
- rotate_blue_block_right: rotate the blue block to the right.
- rotate_blue_block_left: rotate the blue block to the left.
- rotate_pink_block_right: rotate the pink block to the right.
- rotate_pink_block_left: rotate the pink block to the left.
- push_red_block_right: push the red block to the right.
- push_red_block_left: push the red block to the left.
- push_blue_block_right: push the blue block to the right.
- push_blue_block_left: push the blue block to the left.
- push_pink_block_right: push the pink block to the right.
- push_pink_block_left: push the pink block to the left.
- move_slider_left: move the slider to the left.
- move_slider_right: move the slider to the right.
- open_drawer: open the drawer.
- close_drawer: close the drawer.
- turn_on_lightbulb: turn on the lightbulb.
- turn_off_lightbulb: turn off the lightbulb.
- turn_on_led: turn on the led.
- turn_off_led: turn off the led.

**Example Task:**
Now, you will help to parse the goal predicate and generate a sequence of actions to accomplish this task.
Goal predicate: (is-turned-off led)
Symbolic state: is-turned-on(led), is-turned-on(lightbulb), not(is-turned-off(led)), not(is-turned-off(lightbulb))
Task output:
```python
['turn_off_led']
```

**Example Task:**
Goal predicate: (is-turned-on led)
Symbolic state: is-turned-on(led), is-turned-on(lightbulb), not(is-turned-off(led)), not(is-turned-off(lightbulb))
Task output:
```python
[]
```

**Example Task:**
Goal predicate: (is-in red_block drawer)
Symbolic state: not(is-in(red_block, drawer)), not(is-in(red_block, slider)), is-on(red_block, table), not(is-open(drawer)), is-close(drawer), is-slider-left(slider), not(is-slider-right(slider)), not(lifted(red_block))
Task output:
```python
['open_drawer', 'lift_red_block_table', 'place_in_drawer']
```

**Example Task:**
Goal predicate: (is-in red_block drawer)
Symbolic state: not(is-in(red_block, drawer)), not(is-in(red_block, slider)), not(is-on(red_block, table)), is-open(drawer), not(is-close(drawer)), is-slider-left(slider), not(is-slider-right(slider)), lifted(red_block)
Task output:
```python
['place_in_drawer']
```

**Example Task:**
Goal predicate: (and (is-turned-on lightbulb) (is-slider-right slider))
Symbolic state: is-slider-left(slider), not(is-slider-right(slider)), is-turned-off(lightbulb), not(is-turned-on(lightbulb))
Task output:
```python

```
['turn_on_lightbulb', 'move_slider_right']
```

**Additional Instructions:**
1. Make sure the generated plan is a list of actions. Place the list between ```python and
ends with ```.
2. Think Step-by-Step.

