# OpenReview forum: "Learning Compositional Behaviors from Demonstration and Language"
_robot-learning.org/CoRL/2024/Conference — CoRL 2024_

### Official Review · Reviewer_UF9m · 2024-07-21
**Interesting approach and experimental results, but seems like some input assumptions limit how easy it is to apply to a wide-range of tasks**

**Originality:** 3
**Technical Quality:** 4
**Clarity Of Presentation:** 4
**Potential Impact:** 3
**Recommendation:** 3
**Confidence:** 4

**Review:**

Overall, I find this approach of using LLMs to learn skills and grounded predicates interesting and relatively novel, and the demonstrations on robot hardware are impressive (showing generalization across unseen initiation conditions and dealing with partial observability). However, I worry that the autonomy of the method is relatively limited by some of the assumptions, such as needing to have the user explicitly provide the names of predicates, and the contact-based primitives that are used in the body of the operators, which may make it hard to apply this method to tasks in a more general way.

Pros
- The paper is very clear and easy-to-understand, and is described in sufficient detail that I understand the core parts of the algorithm, which is great. I also find the proposed method for using LLMs to generate operators of the skills based on the language annotation and contact sequences to be interesting, and the automatic predicate learning based on the segmentations is nice.

Cons
- “In its current implementation, BLADE requires humans to additionally provide a list of predicate names in natural language”. This seems like a relatively large limitation since the quality of the learned precondition and effects depends largely on the quality of the input predicate names, which may be user dependent and hard for non-experts. The authors also claim that what differentiates them from prior work is that the related works “require manual definitions of planning knowledge”, but this requirement is similar in nature to that, although I understand their assumptions are more general than the assumptions made in these related works, which is good.
- The authors should include / discuss related works of algorithms that autonomously learn symbols / predicates that describe the preconditions and effects of given a skill library without needing human intervention and only requiring skill execution traces, such as [1,2]
- For figure 5, it would be helpful to see what the predicates in these domains are like.

[1] Konidaris, George, Leslie Pack Kaelbling, and Tomas Lozano-Perez. "From skills to symbols: Learning symbolic representations for abstract high-level planning." Journal of Artificial Intelligence Research 61 (2018): 215-289.
[2] Rosen, Eric, et al. "Synthesizing navigation abstractions for planning with portable manipulation skills." Conference on Robot Learning. PMLR, 2023.

**Quality Of The Limitations Section:**

3

**Questions For Rebuttal:**

- “To facilitate LLM generation, we provide additional information on the list of objects with which the robot has contact”. How was this additional information provided, by a human user?
- So all the inferred effects from the LLM are based on the contact sequences and language-annotation with the skill? Would this make it difficult to learn effects that aren’t easily captured by the described contact-based primitives / doesn’t it put a lot of the burden on making sure the language annotation of the skill captures the intended effect?
- “After LLM predicts the definition for all behavior, we will re-segment the demonstrations associated with each behavior based on the LLM-predicted body section”. Why is this done? If the LLM is fed the possible sequence of contact primitives associated with the language description from the dataset, why does the LLM need to predict the body (does it really differ from the input?) / why is this re-segmentation done?
- On line 128, it is stated one of the inputs to the LLM is “a general description of the environment”. Where does this description come from?
- On line 133, it references Fig 3c, but shouldn’t it be referencing 3d?
- For classifier learning: “only pixels associated with the object faucet will be the input to the turned-on(faucet) classifier”. Does this imply the method assumes there is one instance of each category (i.e: there is a single faucet in the scene). If not, if there are multiple instances of the same category (i.e: multiple faucets), how does it associate the pixels with the correct instance?

**Robotics Focus:**

4

**Summary Of Paper:**

This paper proposes BLADE (Behavior from Language and Demonstration), an approach to constructing a skill library and set of high-level action representations that capture the precondition and effects of the skills using LLMs. The input is a set of demonstrations with language descriptions, and the output is a set of operators defining the precondition and effects along with learned grounded classifiers for the predicates, which can then be used by a bi-level planner to execute a novel task.

**Summary Of Recommendation:**

I am quite borderline on this paper, with a lean towards accepting this paper due to the clarity of the presentation, novelty and experimental results, but mostly conditioned on the authors properly addressing the questions raised, and mitigating my concerns over how dependent this method is on the quality of the input predicate names from the user.

---

### Official Review · Reviewer_cYEP · 2024-07-25

**Originality:** 2
**Technical Quality:** 2
**Clarity Of Presentation:** 2
**Potential Impact:** 2
**Recommendation:** 2
**Confidence:** 4

**Review:**

Strength:
The paper introduces BLADE, a framework for long-horizon robotic manipulation that integrates imitation learning with model-based planning. The approach utilizes LLMs to propose abstract behavior descriptions with preconditions, then trains on these predicates via imitation learning. This framework demonstrates improved generalization and replanning capabilities due to geometric constraints.

Weakness:
- A significant issue with using LLMs to propose behavior descriptions, PDDL, and preconditions is their effectiveness in grounding scene context. For example, if actions require minor adjustments or more precise movements, can LLMs accurately ground that information and generate appropriate behavior predicates and preconditions?
- Instead of training classifier models, which may lack generalizability, why not use a vision-language model (VLM)? Using VLMs for components currently utilizing LLMs, as demonstrated in the paper "Manipulate-Anything," could enhance performance.
- The choice of SayCan as a baseline is questionable, especially with a performance score of 0. Considering SayCan's predicates are manually defined, a more suitable baseline might be approaches like "Scaling Up and Distilling Down" or "GPT-4V(ision) for Robotics: Multimodal Task Planning from Human Demonstration."
-Lastly, for those unseen initial conditions, it seems they are still part of the original demonstrations of objects and action sequences. A 3D-based behavior cloning (BC) approach might be more robust to these changes, or increased data augmentation could also address this issue. Developing new behaviors that can be composed of two different predicates or generalized to new initial poses with unseen objects would be a more effective solution.

1.Duan, Jiafei, Wentao Yuan, Wilbert Pumacay, Yi Ru Wang, Kiana Ehsani, Dieter Fox, and Ranjay Krishna. "Manipulate-Anything: Automating Real-World Robots using Vision-Language Models." arXiv preprint arXiv:2406.18915 (2024).

2.Ha, Huy, Pete Florence, and Shuran Song. "Scaling up and distilling down: Language-guided robot skill acquisition." In Conference on Robot Learning, pp. 3766-3777. PMLR, 2023.

3.Wake, Naoki, Atsushi Kanehira, Kazuhiro Sasabuchi, Jun Takamatsu, and Katsushi Ikeuchi. "Gpt-4v (ision) for robotics: Multimodal task planning from human demonstration." arXiv preprint arXiv:2311.12015 (2023).

**Quality Of The Limitations Section:**

3

**Questions For Rebuttal:**

Refer to the reviews

**Robotics Focus:**

4

**Summary Of Paper:**

The paper proposed a framework for long-horizon robotic manipulation by combining imitation learning and model-based planning

**Summary Of Recommendation:**

This is a good work in pushing the field, however there are some design choices i would hope that the author could address, and i am open to changing my rating.

---

### Official Review · Reviewer_ZoiQ · 2024-07-26

**Originality:** 2
**Technical Quality:** 4
**Clarity Of Presentation:** 4
**Potential Impact:** 2
**Recommendation:** 3
**Confidence:** 5

**Review:**

Authors introduce Behavior from Language and Demonstration (BLADE), a framework that integrates imitation learning and model-based planning for long-horizon robotic manipulation. BLADE is leveraging language-annotated demonstrations and extracts abstract action knowledge from large language models (LLMs). BLADE constructs a library of structured, high-level action representations that include preconditions and effects and utilizes (learned) classifiers for evaluating predicates.

Strengths:

-Authors show significant improvement to selected baseline approaches in terms of robustness to disturbances, partial observability, dealing with geometric constraints and dealing with unseen initial states.

BLADE is robust to disturbances as they replan after each action and have symbolic preconditions for actions. It deals with partial observability, by assuming the desired state.  Deals with geometric constraints (e.g. blocked doors) as they are modeled in the precondition to the following actions (e.g. open door). It is generalizing to unseen initial states by Composing learned behaviors, using a model-based planner.

Weaknesses:

-The major concern is that the work relies a lot on the symbolic model. The big part of the model is assumed to be given/generated by LLM (not so realistic). There is no mention of which planner is used, although the planner gives the generalization to unseen states and robustness to the disturbances.

-Although the method is demonstrated on 3 simulated tasks from CALVIN and some real-world examples. Seems that the problems are quite adjusted to the method.

-The most interesting part is the abstract state classifier.  The related automatic predicate annotation is an interesting concept and worth a more elaborate presentation in the paper. They provide most of the generalization capabilities.

-The assumption that user provides the list of predicates in the natural language seems unreasonable. This can bias a lot LLM to generate appropriate operators. There should be more ablation study giving distractor predicates or predicates with non-informative names.

-Segmentation is not clearly explained, most details are at the end in the limitation section.

-As work relies a lot on the symbolic model and model-based planning, it is unclear why it is not compared to the representative approaches in TAMP, such as [1]

[1] Silver, T., Chitnis, R., Kumar, N., McClinton, W., Lozano-Pérez, T.,
Kaelbling, L., & Tenenbaum, J. B. (2023, June). Predicate invention
for bilevel planning. In *Proceedings of the AAAI Conference on Artificial Intelligence* (Vol. 37, No. 10, pp. 12120-12129).

**Quality Of The Limitations Section:**

3

**Questions For Rebuttal:**

-Which planner is used in the work?

-Why is SayCan performance so bad? Which grounding affordance policy is used?

-Are there any negative examples from the predicate annotation? How are they generated?

-How do you get reliable operators with preconditions and effects? It is stated that LLM automatically proposes them from the primitive sequences and language-paired demonstrations. However, it automatic predicate annotation it is assumed to have preconditions and effects to label the predicates.

-How is segmentation done?

**Robotics Focus:**

4

**Summary Of Paper:**

Learns symbolic predicates and predonditions/effects of skills to compose and replan.

**Summary Of Recommendation:**

Interesting paper, but seems incremental and stretching a lot results without clear step forward. Missing some major works in this direction.

---

### Decision · Program_Chairs · 2024-09-04

**Decision:**

Accept

**Comment:**

The paper's strengths include:
BLADE integrates imitation learning and model-based planning, leveraging LLMs to generate symbolic action representations, which is seen as an interesting and relatively novel method.
The framework demonstrates robustness to disturbances, partial observability, and generalization to unseen initial states, making it promising for complex robotic manipulation tasks.
The paper is well-organized and clear, allowing readers to understand the core aspects of the proposed method easily. The experimental results, particularly those involving real-world robots, are impressive and well-documented.
The experiments show significant improvement over baseline approaches, demonstrating the effectiveness of BLADE in various scenarios, including handling geometric constraints and unseen conditions.
The paper's weaknesses include:
The method relies heavily on symbolic models generated by LLMs, which raises concerns about the realism and practicality of these assumptions. The approach's effectiveness is contingent on high-quality input from LLMs, which may not always be feasible.
The requirement for users to provide a list of predicate names in natural language is seen as a significant limitation. This dependence on user input could make the system less autonomous and more challenging for non-experts to use effectively.
The paper does not compare its approach to other representative methods in Task and Motion Planning (TAMP), such as those by Silver et al. (2023), which could provide a more comprehensive evaluation of its contributions.
Certain technical aspects, such as the specifics of the planner used, segmentation, and how reliable operators with preconditions and effects are derived, are not thoroughly explained, leaving some doubts about the reproducibility and robustness of the method.
The choice of SayCan as a baseline is criticized, particularly given its poor performance in the experiments. Reviewers suggest that more relevant and recent approaches could provide a better benchmark for comparison.
While the framework shows potential, its applicability to a broader range of tasks is questioned, especially given its reliance on specific assumptions and symbolic models that may not generalize well to more varied or complex tasks.

Post rebuttal meta-review:
The paper introduces BLADE, a framework that integrates imitation learning and model-based planning for long-horizon robotic manipulation. It leverages language-annotated demonstrations and large language models (LLMs) to create structured action representations.

BLADE constructs a library of high-level action representations, including preconditions and effects grounded in visual perception. The framework is noted for its ability to generalize to novel situations, handle disturbances, and work with partial observability, which are validated both in simulations and on real robots. The approach is relatively novel in integrating LLMs for symbolic action representations in robotic manipulation tasks.

BLADE demonstrates robustness to disturbances and partial observability. The method generalizes well to unseen initial states and novel goals. The framework is well-documented with impressive experimental results, showing significant improvements over baseline methods. Weaknesses:

The reliance on symbolic models generated by LLMs raises concerns about realism and practicality. The method's effectiveness depends heavily on the quality of input from LLMs. The paper lacks a comprehensive comparison with other representative methods in Task and Motion Planning (TAMP), which affects the evaluation of its contributions. Certain technical aspects, like the specifics of the planner and the derivation of operators with preconditions and effects, are not thoroughly explained. Final Recommendation:

Borderline Accept: The paper has strong merits in its novel approach and experimental validation but has significant limitations, particularly in its reliance on LLMs and lack of thorough comparison with other methods. Two reviewers lean towards waek accept with minor reservations, while another expresses concerns about the limitations, suggesting a borderline reject depending on the perspective.